# Towards a Holistic Understanding of Selection Bias for Causal Effect Identification

Yiwen Qiu [1]   Filip Kovačević [1]   Shimeng Huang [1]   Peter Spirtes [2]   Francesco Locatello [1]

## Abstract

Selection bias is pervasive in observational studies. For example, large scale biobanks data can exhibit "healthy volunteer bias" when respondents are healthier and of higher socio-economic status than the population they are meant to represent (Swanson, 2012; van Alten et al., 2024). Recovering causal effects from such sub-population is an important problem in causal inference, as estimating average treatment effects (ATE) from selected populations can result in a severely biased estimate of the ATE from the whole population. In this paper, we investigate the identifiability of the ATE under selection bias. We provide *necessary and sufficient conditions* for ATE identifiability, leveraging assumptions on probability classes to characterize propensity score and selection probability, which are weaker assumptions than those required by existing general identification frameworks . Compared to previous works, our results extend existing graphical identifiability criteria and offer a more comprehensive understanding of causal effect identification *with strictly weaker conditions* in the presence of selection bias.

## 1. Introduction

In many real-world scenarios, the data we collect is often subject to selection bias, which arises from the preferential inclusion of certain data points that are dependent on some of the observed variables (Heckman, 1979). This bias poses a significant challenge in statistical inference. For instance, in medical studies, patients who volunteer for clinical trials may differ systematically from those who do not, leading to biased estimates of treatment effects (Hernán et al., 2004). In social sciences, survey respondents may not represent the broader population due to non-random participation, affect-

ing the validity of conclusions drawn from such data (Stone et al., 2024). More concretely, we can consider the following Example 1.1 (illustrated in Figure 1).

*Example* 1.1 (Biased ATE estimation). The government wants to estimate the ATE of a physical activity subsidy $T$ (e.g. opening a free community fitness center) on overall cardiovascular health ($Y$) for the entire country. Data is collected via a voluntary digital health survey. Because the survey is intensive and digital-only, the respondents (the subpopulation $S = 1$) are primarily individuals with higher socio-economic status (SES) and higher education levels. We use $Y(1)$ to denote the potential outcome of $Y$ if one receives the subsidy ($T = 1$), and $Y(0)$ otherwise.

In this case, there is a discrepancy between the ATE in the selected subpopulation ($\text{ATE}^{\text{obs}}$) and that in the whole population ($\text{ATE}^{\text{all}}$). For high-SES individuals, the subsidy might have a smaller marginal effect. They likely already exercise or have the means to stay healthy. Adding a free fitness center might only result in a small improvement in cardiovascular health. In contrast, for low-SES individuals, the subsidy could have a more substantial impact, as they might have fewer opportunities for physical activity due to cost or neighborhood safety.

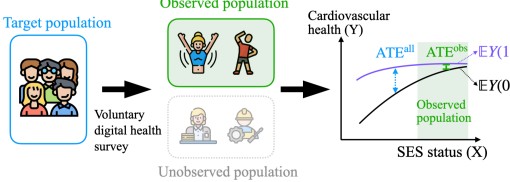

*Figure 1.* Illustration of selection bias in estimating the ATE of a physical activity subsidy ($T$) on cardiovascular health ($Y$). The participation into the survey is influenced by SES, which creates a selection bias that can lead to incorrect estimates of the ATE ($\text{ATE}^{\text{obs}} \ll \text{ATE}^{\text{all}}$) if not properly accounted for.

This work aims to address the challenges posed by selection bias in causal inference. We propose a novel framework that is general enough to encompass various existing models while being specific enough to provide clear identifiability results. We make the following contributions:

**General definition of selection bias.** We provide a general definition of selection bias, not only in terms of M-bias

---

[1]Institute of Science and Technology Austria (ISTA) [2]Carnegie Mellon University. Correspondence to: Yiwen Qiu <yiwen.qiu@ista.ac.at>.

*Proceedings of the 43^rd International Conference on Machine Learning*, Seoul, South Korea. PMLR 306, 2026. Copyright 2026 by the author(s).

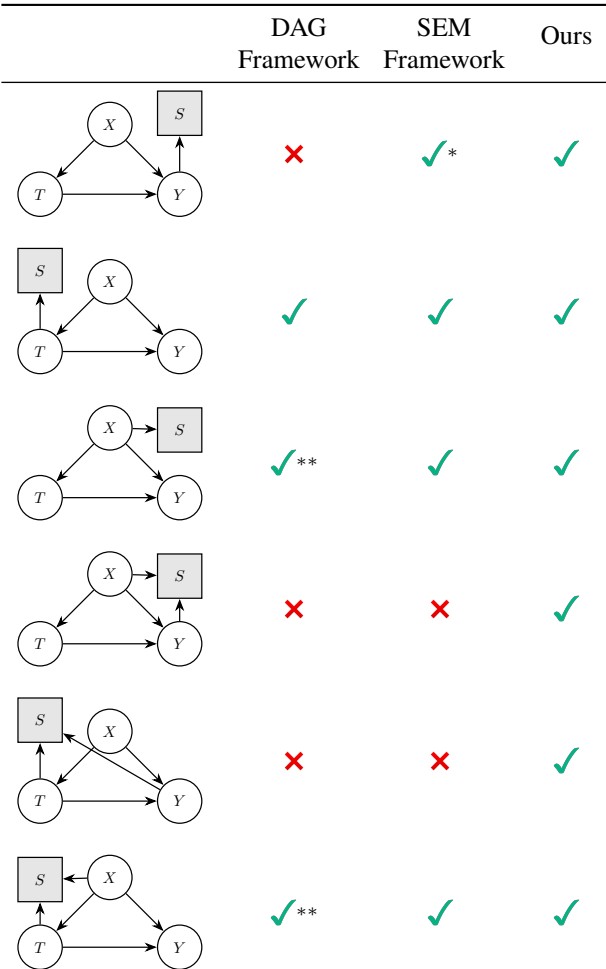

| | DAG Framework | SEM Framework | Ours |
|---|:---:|:---:|:---:|
| | ✗ | ✓* | ✓ |
| | ✓ | ✓ | ✓ |
| | ✓** | ✓ | ✓ |
| | ✗ | ✗ | ✓ |
| | ✗ | ✗ | ✓ |
| | ✓** | ✓ | ✓ |

*Table 1.* A comparison of different frameworks. The DAG Framework refers to the result in (Correa et al., 2018), which gives identifiability results assuming common DAG assumptions (Definition 3.7, Definition 3.8) hold. The SEM Framework refers to the result in (Zhang et al., 2016), which assumes an additive noise model (ANM) and non-Gaussian noise. With these additional parametric assumptions, they are able to identify causal effect when the selection is only outcome-dependent (Zhang et al., 2016). In the chart, ✓ indicates that the causal effect in the *whole* population ( 2.2) is identifiable under the framework, provided with the assumptions in ∗(ANM model and non-Gaussian noise) and ∗∗(access to external unbiased distribution on the covariate), while ✗ indicates that it is not identifiable or not considered. Our identifiability results cover all the scenarios which are identifiable through other frameworks, and extend the results to scenarios which are not considered identifiable before. For a comparison of causal effect identifiability results in sub-populations, please refer to Appendix C.

or collider stratification bias (Greenland, 2003; Elwert & Winship, 2014). Selection bias is most widely understood in epidemiological studies as collider stratification bias (Greenland, 2003; Hernán & Robins, 2010; Elwert & Winship, 2014), where conditioning on a collider variable (or its descendants) can induce spurious associations between its causes. However, selection bias can manifest in

other forms, not necessarily due to the collider effect, e.g., manifest itself in various graphical forms (Smith, 2020; Lu et al., 2022). We gave the general definition of selection bias in Definition 2.1.

**General solution.** We do not need to detect the location where selection occurs, in contrast to the various works based on graphical models (Mathur & Shpitser, 2025a; Jaber et al., 2018; Zhang & Lu, 2025; Mathur & Shpitser, 2025b; Perković et al.; Bareinboim & Pearl, 2012; Bareinboim et al., 2014; Bareinboim & Tian, 2015; Bareinboim et al., 2022). Table. 3 compares our framework with existing works. Previous works often rely on identifying the exact location of selection bias in the causal directed acyclic graph (DAG), which can be challenging in practice. Our approach circumvents this requirement by providing a general solution that does not depend on pinpointing the selection mechanism. For a complete literature review, please refer to Appendix A.

**Identifiability under weaker assumptions.** Apart from restrictive assumptions on where the selection happens, it is also common for existing works to have strong distributional assumptions or specific functional forms, such as non-Gaussian noise and additive noise models (ANM) for causal effect identifiability under selection bias[1]. Our framework imposes mild assumptions firstly by allowing selection to depend on arbitrary observed variables, and secondly by requiring weaker distributional assumptions. We achieve this through leveraging recent advances in the truncated statistics (Cohen Jr, 1950; Daskalakis et al., 2021; Kontonis et al., 2019; Lee et al., 2024), which allow us to perform extrapolation from given samples from their truncations to their whole distribution.

## 2. Notations and Definitions

We consider a general potential outcomes model. An observational study involves units (*e.g.*, patients) with covariates $X \in \mathbb{R}^d$ (*e.g.*, medical history). Each unit receives a binary treatment $T \in \{0, 1\}$ (*e.g.*, medication) with a fixed but unknown probability, independent across units, and we observe a treatment-dependent outcome $Y(T) \in \mathbb{R}$ (*e.g.*, symptom severity). The tuple $(X, Y(0), Y(1), T)$ follows an unknown joint distribution $\mathcal{D}$, which defines the target study. For each $t \in \{0, 1\}$, $\mathcal{P}_{X,Y(t)}$ denotes the marginal distribution of $X$ and $Y(t)$ and $\mathcal{P}_X$ the marginal distribution of $X$. To simplify the exposition, we assume that $\mathcal{P}_{X,Y(0)}$ and $\mathcal{P}_{X,Y(1)}$ are continuous distributions with densities throughout. The goal is to estimate the average treatment effect (ATE) over the target study's population , defined as $\tau_{\mathcal{P}} := \mathbb{E}_{\mathcal{P}}[Y(1) - Y(0)]$.

We only have access to the observational study $\mathcal{P}$ of

---

[1]Note that, these works provide the sufficient condition, i.e. the causal effect is not identifiable by the provided method (backdoor adjustment etc.) but might still be identifiable with other methods.

$(X, Y(T), T)$ because each individual only receives a specific treatment, and we do not observe the counterfactuals. Without further assumptions, ATE is not identifiable from $\mathcal{D}$ (Imbens & Rubin, 2015), and the identifiability problem is even harder in the presence of selection. To model selection procedures, we define the set of variables as $\mathbf{V} = \{X, Y, T\}$, and $S$ is the binary selection variable indicating the inclusion of data point in the dataset, i.e. we only observe $\mathcal{P}(\mathbf{V} \mid S = 1) \coloneqq \mathcal{P}^S$.

In order to specify the conditions for ATE identifiability, we define three distributions: the *propensity scores* $P_{t|x}(t, x) \coloneqq P_{t|x} = \Pr(T = t \mid X = x)$ for each $t \in \{0, 1\}$, the joint probability of covariate-outcome $P_{xy(t)}(x, y, t) \coloneqq P_{xy(t)} = \Pr(X = x, Y(t) = y)$, and the *selection probability* $P_{s|xyt}(x, y, t) = \Pr(S = 1 \mid X = x, Y(t) = y, T = t)$. We say that a tuple $(P_{t|x}, P_{xy(t)}, P_{s|xyt})$ is *compatible* if it defines a valid distribution over $(X, T, Y(0), Y(1), S)$.

Correspondingly, we construct the following distribution classes: the class of propensity scores $\mathbb{P}_{t|x} \subseteq \{p \colon \{0, 1\} \times \mathbb{R}^d \to [0, 1]\}$, the class of covariate-outcome distributions $\mathbb{P}_{xy(t)} \subseteq \Delta(\mathbb{R}^d \times \mathbb{R}) \times \{0, 1\}$, and the class of selection distributions $\mathbb{S} \subseteq \{p \colon \mathbb{R}^d \times \mathbb{R} \times \{0, 1\} \to [0, 1]\}$. We denote the full space of $(X, Y, T)$ as $\mathcal{A} \coloneqq \mathbb{R}^d \times \mathbb{R} \times \{0, 1\}$, and the region that are completely censored as $\mathcal{B} \subseteq \mathcal{A}$ where $P_{s|xyt} = 0$ on $\mathcal{B}$. We say that a tuple $(P_{t|x}, P_{xy(t)}, P_{s|xyt})$ is compatible when it defines a valid distribution, and we say that an observational study $\mathcal{P}$ is *realizable* with respect to the distribution class tuple $(\mathbb{P}_{t|x}, \mathbb{P}_{xy(t)}, \mathbb{P}_{s|xy(t)t})$ when the induced probabilities satisfy $P_{t|x} \in \mathbb{P}_{t|x}$, $P_{xy(t)} \in \mathbb{P}_{xy(t)}$ and $P_{s|xyt} \in \mathbb{P}_{s|xy(t)t}$. See Appendix B.1 for formal definitions of *compatibility* and *realizability*.

**Definition 2.1** (Selection Bias). Selection bias refers to preferential inclusion of units from the sample so that $\mathcal{P}(\mathbf{V}|S = 1) \neq \mathcal{P}(\mathbf{V})$, where $S$ is a binary variable indicating the membership of a unit in the dataset.

In this paper, we care about the question of whether ATE is identifiable from the selected population distribution $\mathcal{P}(\mathbf{V} \mid S = 1)$. We define two notions of ATE identifiability under selection bias (s-ATE) as follows:

**Definition 2.2** (s-ATE-full: identifiablity in full population). We say that an ATE $\tau_{\mathcal{P}}$ is *identifiable* from the distribution $\mathcal{P}(\mathbf{V} \mid S)$, if there is a mapping $f$ such that $f(\mathcal{P}_{V|S}) = \tau_{\mathcal{P}}$ for any observational study $\mathcal{P}$ realizable with respect to such $(\mathbb{P}_{t|x}, \mathbb{P}_{xy(t)}, \mathbb{P}_{s|xy(t)t})$.

**Definition 2.3** (s-ATE-sub: identifiablity in sub-population). We say that an ATE $\tau_{\mathcal{P}^S}$ is *identifiable* from the distribution $\mathcal{P}(\mathbf{V} \mid S)$, if there is a mapping $f$ such that $f(\mathcal{P}_{V|S}) = \tau_{\mathcal{P}^S}$ for any observational study $\mathcal{P}$ realizable with respect to such $(\mathbb{P}_{t|x}, \mathbb{P}_{xy(t)}, \mathbb{P}_{s|xy(t)t})$.

*Discussion* 2.4 (Difference between s-ATE-full and s-ATE–sub). This paper mostly focuses on s-ATE-full, which is more challenging because the estimand concerns partly unobserved data. Definition 2.2 essentially states that if two selected populations are identical $\mathcal{P}^1(\mathbf{V} \mid S) = \mathcal{P}^2(\mathbf{V} \mid S)$, then the ATE is also identical in the full population ($\tau_{\mathcal{P}^1} = \tau_{\mathcal{P}^2}$). Although this is not true in general, we show that it holds when a *certain* condition on the distribution class triple $(\mathbb{P}_{t|x}, \mathbb{P}_{xy(t)}, \mathbb{P}_{s|xy(t)t})$ is met (Condition 1). On the other hand, Definition 2.3 focuses on identifying the ATE within the selected sub-population ($\tau_{\mathcal{P}^1_{V|S}} = \tau_{\mathcal{P}^2_{V|S}}$). Two identifiability notions are needed in different practical scenarios, especially when the ATE in the full population is different from the selected sub-population. For instance, in public health policy making, the government wants to estimate the average improvement in health conditions for the entire city, not just one neighborhood, then Definition 2.2 becomes more relevant; while in pharmaceutical research, a cancer drug might be ineffective for the general population but highly effective for patients who has entered the research - the sub-population that has diagnosed with cancer (Definition 2.3).

Note that it is far more challenging to estimate treatment effects under selection bias compared to estimating the conditional average treatment effect (CATE) (Imbens & Rubin, 2015). While CATE is also targeted at ATE on a sub-population, which makes it superficially seem similar to s-ATE-sub, for CATE, one can simply condition on the covariates $X$, i.e., $\tau_{\mathcal{P}}(x) \coloneqq \mathbb{E}_{\mathcal{P}}[Y(1) - Y(0) \mid X = x]$. Under selection bias, we generally lack the knowledge of the selection procedure $P(S = 1 \mid \mathbf{V})$, which may depend not only on the covariates but also on treatment or the outcome of interest. This means estimating the s-ATE-full is even more complex, requiring further assumptions to extrapolate beyond the observed population.

## 3. ATE Identifiability Conditions

In general, ATE is not identifiable even in the absence of selection bias, i.e. it cannot be uniquely expressed in terms of the observed data distribution unless we make additional assumptions (Rosenbaum & Rubin, 1983; Imbens & Rubin, 2015) like unconfoundedness (ignorability), positivity (overlap), Stable Unit Treatment Value Assumption (SUTVA), and consistency (See Appendix B.1 for more details), but no work has been developed to identify ATE under general selection bias, limiting to specific cases like (Zhang et al., 2016; Bareinboim & Pearl, 2011; Bareinboim & Tian, 2015).

First (Section. 3.1), we present our main identifiability conditions for ATE with selection bias (Condition 1). Then in Section. 3.2, the condition is accompanied by propositions that instantiate the conditions for specific distribution classes, i.e. restrictions on $\mathbb{P}_{t|x}, \mathbb{P}_{xy(t)}$ (and $\mathbb{S}$), such that the conditions hold. In particular, Proposition 3.3 and Proposition 3.5 are of practical interest as they instantiate Con-

dition 1 for deterministic and non-deterministic selection mechanisms, respectively, and we find that ATE is identifiable under common distribution families under selection. This is an exciting result, extending the identifiability of ATE under selection to settings that were previously considered unidentifiable in the literature (Zhang et al., 2016; Correa & Bareinboim, 2017; Correa et al., 2019). All the proofs for the theorem and propositions are provided in the appendix.

### 3.1. An Unified Framework for ATE Identifiability

In this section, we present identifiability conditions for ATE in terms of restrictions on the classes of propensity scores $\mathbb{P}_{t|x}$ and covariate-outcome distributions $\mathbb{P}_{xy(t)}$, which serves as an unified framework to characterize the specific distributions that satisfy this condition later (Section 3.2)

---

**Condition 1** (Identifiability Condition under Selection). The distribution classes $(\mathbb{P}_{t|x}, \mathbb{P}_{xy(t)}, \mathbb{S})$ satisfy the *Identifiability Condition under Selection* if for any tuples $(P_{t|x}, P_{xy(t)}, P_{s|xyt}), (Q_{t|x}, Q_{xy(t)}, Q_{s|xyt}) \in \mathbb{P}_{t|x} \times \mathbb{P}_{xy(t)} \times \mathbb{S}$ that are compatible with $(\mathbb{P}_{t|x}, \mathbb{P}_{xy(t)}, \mathbb{S})$, it holds that

$$\underbrace{\tau_{P_{xy(t)}} \neq \tau_{Q_{xy(t)}}}_{①} \implies \exists (x, y, t) \in \mathcal{A}, \text{ s.t. } ② \text{ is valid,}$$

$$\text{for } \underbrace{\alpha_P(x, y, t) P_{t|x} P_{xy(t)} \neq \alpha_Q(x, y, t) Q_{t|x} Q_{xy(t)}}_{②},$$

where $\alpha_P := \frac{P_{s|xyt}}{P(s)}, \alpha_Q := \frac{Q_{s|xyt}}{Q(s)}$.

---

Note that, only the distribution product that are marked in brown color in Condition 1 are observed, and it corresponds to the observed distribution $P(X, Y, T \mid S = 1)$. Without further information, we do not have access to the unbiased marginal $P_X$ and $Q_X$ thus cannot test ②. When we do have such access to the unbiased marginal, as in the cases in BioBank data where we have a matching population (Swanson, 2012; van Alten et al., 2024), we provided the augmented Condition 2 in Appendix B.2. The Condition 2 is less restrictive in the sense that it allows a wider range of distributions that satisfy it, while requiring more information as input. This condition is related to that of (Cai et al., 2025), which provides a distributional condition for the identifiability of the ATE without selection bias, but allowing for overlap violations. This is a fundamentally different setting, as overlap violations are a challenge arising from the support of the data's true underlying distribution, while selection bias is a distortion of a causal effect stemming from how data enters the analysis. This is reflected in our definition with the inclusion of the selection mech-

anism. Concretely, this means that the condition in (Cai et al., 2025) can be used to identify the s-ATE-sub (Definition 2.3) but not s-ATE-full (Definition 2.2), which is the more challenging scenario that this paper focuses on.

In this paper, we focus on the cases where there is no unobserved confounding, i.e. $Y(t) \perp\!\!\!\perp T \mid X$ for $t \in \{0, 1\}$, and we further assume overlap in the unselected population, a common assumption in causal inference. We characterize the class of *c-overlap* and *unconfounded* distributions by restricting the propensity scores class $\mathbb{P}_{t|x}$ as follows:

$$\mathbb{P}_{t|x}(c) := \{p : \{0, 1\} \times \mathbb{R}^d \to [0, 1] \mid c < p(t, x) < 1 - c\}.$$

After selection, however, the observed distribution may no longer satisfy overlap. In this work, we consider two types of selection mechanisms: deterministic and nondeterministic selection. The class of *deterministic-selection* distributions on $\mathcal{B} \subseteq \mathcal{A}$ is defined as:

$$\mathbb{S}^{\text{det}}(\mathcal{B}) := \{p : \mathcal{A} \to [0, 1] \mid p(x, y, t) = 0, \forall (x, y, t) \in \mathcal{B}\},$$

For nondeterministic selection, we define the class of *d-selection-overlap* distributions on $\mathcal{B} \subseteq \mathcal{A}$ (with a slight overload of notation) as:

$$\mathbb{S}(\mathcal{B}, d) := \{p : \mathcal{A} \to [0, 1] \mid p(x, y, t) > d, \forall (x, y, t) \in \mathcal{B}\},$$

and denote by $\mathbb{S}(d) := \mathbb{S}(\mathcal{A}, d)$ where $\mathcal{A} = \mathbb{R}^d \times \mathbb{R} \times \{0, 1\}$. For deterministic selection, we define $\mathbb{S}_1^{\text{det}}$ by considering any deterministic selection mechanism, i.e.

$$\mathbb{S}_1^{\text{det}} := \{p : \mathcal{A} \to [0, 1] \mid \exists \mathcal{B} \subset \mathcal{A}, \mu(\mathcal{B}^c) > 0, \ p = \mathbb{1}_{\mathcal{A} \setminus \mathcal{B}}\},$$

where $\mathcal{B}^c = \mathcal{A} \setminus \mathcal{B}$, $\mathbb{1}_{\mathcal{A} \setminus \mathcal{B}} = \begin{cases} 1, & \text{for } (x, y, t) \in \mathcal{B}^c, \\ 0, & \text{for } (x, y, t) \in \mathcal{B}, \end{cases}$

and $\mu$ is the product measure of Lebesgue measure and a counting measure.

Given these definitions, we present our main theorem characterizing the identifiability of ATE under selection bias in Theorem 3.1.

**Theorem 3.1** (Characterization of s-ATE-full). *The ATE $\tau_P$ is identifiable from any observational distribution $P$ realizable with respect to $(\mathbb{P}_{t|x}, \mathbb{P}_{xy(t)}, \mathbb{S})$ if and only if $(\mathbb{P}_{t|x}, \mathbb{P}_{xy(t)}, \mathbb{S})$ satisfy Condition. 1.*

Formally, the following hold:

1. **(Sufficiency)** If $(\mathbb{P}_{t|x}(c), \mathbb{P}_{xy(t)}, \mathbb{S})$ satisfies Condition 1, then there is a mapping $f : \Delta(\mathbb{R}^d \times \mathbb{R} \times \{0, 1\}) \to \mathbb{R}$ with $f(P(V \mid S)) = \tau_P$ for each distribution $P(V \mid S)$ realizable with respect to $(\mathbb{P}_{t|x}(c), \mathbb{P}_{xy(t)}, \mathbb{S})$.

2. **(Necessity)** Otherwise, for any map $f : \Delta(\mathbb{R}^d \times \mathbb{R} \times \{0, 1\}) \to \mathbb{R}$, there exists a distribution $P(V \mid S)$ realizable with respect to $(\mathbb{P}_{t|x}(c), \mathbb{P}_{xy(t)}, \mathbb{S})$ such that $f(P) \neq \tau_P$.

*Discussion* 3.2 (Inuition for Condition 1). The key idea behind Condition 1 is to ensure that if two distributions $\mathcal{P}$ and $\mathcal{Q}$ have different ATEs (i.e. ① holds), then they must differ in the observed parts of the distribution (i.e. ② holds) for the ATE to be identifiable. In general, under selection bias, this condition does not hold, as two distributions can have different ATEs while being indistinguishable in the observed data due to the selection mechanism. However, by imposing restrictions on the classes of propensity scores $\mathbb{P}_{t|x}$, covariate-outcome distributions $\mathbb{P}_{xy(t)}$, and selection mechanisms $\mathbb{S}$, we can ensure that Condition 1 holds, leading to the identifiability of ATE as stated in Theorem 3.1. The propositions that follow provide specific instances of these restrictions that guarantee the satisfaction of Condition 1.

## 3.2. Instantiation of Identifiable Scenarios

In Section. 3.1, Theorem 3.1 serves an important structural purpose by providing a unified language, and in this section, we show that this language allows us to provide new identifiability guarantees (Propositions 3.3 and 3.5) and cast all existing graphical identifiability results (Corollaries 3.9–3.13) into a single framework in a coherent manner.

**Proposition 3.3** (Cases under Deterministic Selection). *The tuple* $(\mathbb{P}_{t|x}(c), \mathbb{P}_{xy(t)}^{C^\infty}, \mathbb{S}_1^{det})$ *satisfies Condition 1,* $\forall c,$ $0 < c < \frac{1}{2}$, *where* $\mathbb{P}_{xy(t)}^{C^\infty} \subseteq \Delta(\mathbb{R}^d \times \mathbb{R}) \times \{0,1\}$ *is a family of distributions such that for any* $P_{xy(t)} \in \mathbb{P}_{xy(t)}^{C^\infty}$, *and for any* $t \in \{0,1\}$, *its corresponding conditional distribution* $P_{y(t)|x}$ *is parametrized as* $P_{y(t)|x} \propto e^{f(x,y)}$, *and its corresponding marginal distribution* $P_x$ *is parametrized as* $P_x \propto e^{g(x)}$, *where* $f(x,y) = f_{P_{xy(t)}}$ *and* $g(x) = g_{P_{xy(t)}}$ *are polynomial functions of* $(x,y)$, *and* $x$, *respectively.*

*Discussion* 3.4. When selection is deterministic, ATE is identifiable under c-weak-overlap propensity scores and smooth enough outcome distributions. This is a notable result as it covers a wide range of practical scenarios where Gaussian and other common distributions are assumed. The intuition behind Proposition 3.3 is that under deterministic selection, the observed data corresponds to a truncated version of the original distribution. By assuming that the covariate-outcome distribution is smooth enough (e.g., belonging to $\mathbb{P}_{xy(t)}^{C^\infty}$), we can leverage results from truncated statistics (Daskalakis et al., 2021) to extrapolate and recover the original distribution, leading to ATE identifiability.

**Proposition 3.5** (Cases under Nondeterministic Selection). *The tuple* $(\mathbb{P}_{t|x}(c), \mathbb{P}_{xy(t)}, \mathbb{S}(d))$ *satisfies Condition 1,* $\forall c, 0 < c < \frac{1}{2}$, *if* $\mathbb{P}_{xy(t)}$ *belongs to the following distribution classes: (1) Gaussian distribution, (2) Laplace distribution (light tailed distributions), and (3) Pareto distribution, (4) Log-normal distributions (heavy tailed distributions).*

*Discussion* 3.6. When selection is non-deterministic, ATE is identifiable under c-weak-overlap and a wide range of

distribution families. This extends the identifiability of ATE under selection to settings that were previously considered unidentifiable in the literature (Zhang et al., 2016; Correa & Bareinboim, 2017; Correa et al., 2019), for which the intuition is that for any $\mathcal{P}$ and $\mathcal{Q}$ in these distribution classes that have different means of the outcomes, we can always find a point in the support of the distributions where the observed distributions differ enough, i.e. ② holds in Condition 1.

### 3.3. Connections to Graphical Conditions in Prior Work

In this section, we connect our identifiability conditions to existing graphical criteria for s-ATE-full (Corollary 3.9-3.12) and s-ATE-sub (Corollary 3.14). In particular, we show that our conditions are more general than the backdoor and the selection-backdoor criteria (Correa & Bareinboim, 2017; Correa et al., 2019).

To do this, we assume a causal graph $\mathcal{G}$ over the variables $V = \{X, T, Y, S\}$. The graph $\mathcal{G}$ is Markovian (Definition 3.7) and faithful (Definition 3.8) to the distribution.

**Definition 3.7.** (Markov Condition (Spirtes et al. (2001); Pearl (2009))) Given a DAG $\mathcal{G}$ and distribution $\mathbb{P}$ over the variable set $V$, every variable $X$ in $V$ is probabilistically independent of its non-descendants given its parents in $\mathcal{G}$.

**Definition 3.8.** (Faithfulness Assumption (Spirtes et al. (2001); Pearl (2009))) There are no independencies between variables that are not entailed by the Markov Condition.

We denote by $\mathcal{G}_{\overline{M}}$ the graph obtained from $\mathcal{G}$ by removing all incoming edges to $M$, and by $\mathcal{G}_{\underline{M}}$ the graph obtained from $\mathcal{G}$ by removing all outgoing edges from $M$. We also denote by $Anc(S)$ the set of ancestors of $S$ in $\mathcal{G}$.

**Corollary 3.9** (Selection-backdoor (Correa et al., 2019) as a special case of Condition 1). *If $S$ is a child of $T$ but not a child of $X$ nor $Y$, i.e.* $T \to S, X \nrightarrow S$, *and* $Y \nrightarrow S$ *in $\mathcal{G}$, then (a) $X$ satisfies the selection-backdoor criterion in the causal graph $\mathcal{G}$, and (b) the distribution classes* $(\mathbb{P}_{t|x}, \mathbb{P}_{xy(t)})$ *that are entailed by $\mathcal{G}$ satisfy Condition 1.*

**Corollary 3.10** (Selection-backdoor w/ external unbiased covariate (Correa et al., 2019) as a special case of Condition 1). *If $S$ is a child of $X$ but not a child of $Y$, i.e.* $X \to S$ *and* $Y \nrightarrow S$, *then (a) $X$ satisfies the selection-backdoor-ext criterion in the causal graph $\mathcal{G}$, and (b) the distribution classes* $(\mathbb{P}_{t|x}, \mathbb{P}_{xy(t)}, \mathbb{S})$ *that are entailed by $\mathcal{G}$ satisfy Condition 1.*

*Discussion* 3.11. Corollary 3.9 (and similarly for 3.10) show the cases where the selection backdoor criteria (or with external data) is satisfied by $X$ in the causal graph (statement (a)), and those cases can be interpreted as Condition 1 is satisfied in the distributions that the graph entails (statement (b)).

**Corollary 3.12** (Extension of outcome-dependent selection (Zhang et al., 2016))**.** *If the selection mechanism is outcome-dependent, i.e., there exist $Y \to S$ in $\mathcal{G}$, then the distribution classes $(\mathbb{P}_{t|x}, \mathbb{P}_{xy(t)}, \mathbb{S})$ that are entailed by $\mathcal{G}$ satisfy Condition 1 when they belong to the distribution classes as states in Proposition 3.3 and Proposition 3.5.*

*Discussion* 3.13. Note that Corollary 3.12 extend the existing results in the literature (Zhang et al., 2016), which state that causal effects are only identifiable under outcome-dependent selection mechanisms when the noise is non-Gaussian, modeled by an additive noise model, and also extends the result in (Correa & Bareinboim, 2017; Correa et al., 2019). The main reason is that we incorporate assumptions on the distribution classes $\mathbb{P}_{t|x}(c), \mathbb{P}_{xy(t)}$ which allows us to leverage the result from truncated statistics (Daskalakis et al., 2021) for extrapolation. We remark that these assumptions are actually milder compared to some prior work, e.g. Zhang et al. (2016), which specifically restricts to Gaussian distributions.

**Corollary 3.14** (S-id (Abouei et al., 2024) graphical criteria as a special case of Condition 3)**.** *If $S$ satisfies the* S-id *graphical criteria in the causal graph $\mathcal{G}$, in particular, $T \notin Anc(S)$, then the distribution classes $(\mathbb{P}_{t|x}, \mathbb{P}_{xy(t)}, \mathbb{S})$ that are entailed by $\mathcal{G}$ satisfy Condition 3.*

*Discussion* 3.15. Intuitively, one would think that s-ATE-sub would hold if the original $P(\mathbf{V})$ guarantees ATE identifiability, as long as we take $P(\mathbf{V} \mid S)$ as a new distribution $P'(V)$. However, this is not the case, as selection bias can lead to $T \not\perp\!\!\!\perp Y(t) \mid X$ in distribution $P'(V)$, which violates the unconfoundedness (Assumption B.4) for ATE identifiability.

# 4. Estimation

When the distribution classes satisfy Proposition 3.3 or 3.5, we can estimate s-ATE-full by following a three-stage procedure summarized in Algorithm 1.

First, we identify the region $\mathcal{B}$ where the data is not subject to deterministic censoring, i.e. where the propensity score $\hat{e}(x) = \hat{P}(t = 1|x)$ satisfies $c < \hat{e}(x) < 1 - c$. Data within this region is only subject to non-deterministic selection bias and can be used to estimate the conditional outcome distribution $\hat{P}(y|x, t)$, correcting for the selection bias using the learned selection bias function $\hat{\beta}(x, y, t)$. We describe two methods for estimation: maximum likelihood (Section 4.1) and score matching (Section 4.2). Finally, the ATE is then estimated by sampling from the estimated population distribution and performing counterfactual inference using the learned outcome model.

## 4.1. Maximum likelihood estimation (MLE)

To estimate the parameters $\theta = \{\theta_f, \theta_\beta\}$, we minimize the negative log-likelihood of the observed data within the region $\mathcal{B}$ with common support:

$$
\begin{aligned}
L(\theta) = &- \sum_{i \in \mathcal{D}_\mathcal{B}} \left( \log \hat{P}(x_i, y_i, t_i \mid s = 1) \right) \\
= &- \sum_{i \in \mathcal{D}_\mathcal{B}} \left( \log \hat{P}(y_i|x_i, t_i) + \log \hat{\beta}(x_i, y_i, t_i) \right. \\
& \left. \hat{P}(x_i) + \hat{e}(x_i) \right)
\end{aligned}
\tag{1}
$$

During optimization, the solution to $\hat{P}(y|x, t)$ and $\hat{\beta}(x, y, t)$ may suffer from some indeterminacy, e.g., differ by a constant or goes to $\infty$. To find the solution for which the biased selection procedure is as weak as possible, we regularize the selection function to make sure $\log(\hat{\beta})$ is close to 0. Therefore, the final objective function is

$$
L(\theta) + \lambda \sum_{i \in \mathcal{D}_\mathcal{B}} (\log \|\hat{\beta}(x_i, y_i, t_i)\|_2^2),
$$

where $\lambda$ is a hyperparameter controlling the regularization.

## 4.2. Score Matching

Alternatively, we can estimate the parameters by score matching (Hyvärinen & Dayan, 2005), i.e., by minimizing the expected squared distance between the gradient of the log-density given by the model and the gradient of the log-density of the observed data. This procedure has the advantage that it aims to match the shape of the two densities and is invariant to the scaling factor of the model density.

We define the score function for any random variable $Z$ as $\psi(z; \theta) = (\psi_1(z; \theta), \ldots, \psi_m(z; \theta))^\top = \left( \frac{\partial \log p_Z(z; \theta)}{\partial z_1}, \ldots, \frac{\partial \log p_Z(z; \theta)}{\partial z_m} \right)^\top$. The objective function for score matching is then given by

$$
\begin{aligned}
J(\theta) = \frac{1}{N} \sum_{i \in \mathcal{D}_\mathcal{B}}^N \Big( &\frac{1}{2} \|\psi(x_i, y_i, t_i; \theta)\|^2 \\
&+ \operatorname{tr}(\nabla \psi(x_i, y_i, t_i; \theta)) \Big)
\end{aligned}
\tag{2}
$$

---

**Algorithm 1** ATE Estimation under Selection Bias

---

**input** Samples under selection bias $\mathcal{D} = \{X_i, Y_i, T_i\}_{i=1}^N$.
**output** ATE on the whole population.
 Estimate propensity score $\hat{e}(x) = \hat{P}(t = 1|x)$.
**set** region $\mathcal{B} = \{x \mid c < \hat{e}(x) < 1 - c\}$.
 Filter dataset: $\mathcal{D}_\mathcal{B} \leftarrow \{(x_i, y_i, t_i) \mid x_i \in \mathcal{B}\}$.
 Estimate $\hat{P}(y|x, t)$ either by minimizing the negative log-likelihood for the observed (selected) data $L(\theta)^{\mathrm{MLE}}$, or by score matching and minimizing $J(\theta)$.
 Estimate population density $\hat{P}_{pop}(x)$ by reweighting observed samples $x_i$ with $1/\beta(x_i, y_i, t_i)$.
 Calculate
 $\hat{\tau}_P = \mathbb{E}_{x \sim \hat{P}_{pop}(x)} \left[ \mathbb{E}_{y \sim \hat{P}(y|x, t=1)}[y] - \mathbb{E}_{y \sim \hat{P}(y|x, t=0)}[y] \right]$
 Return $\hat{\tau}_P$.

---

Let $p(y|x)$ be the target unbiased density and $\tilde{p}(y|x)$ be

|  | IPW | Polynomial | MLE | MLE $+\beta$ | Score Matching | Score Matching $+\beta$ |
|---|---|---|---|---|---|---|
| Deterministic Selection | ✗ | ✓ | ✓ | ✓ | ✓ | ✓ |
| Non-deterministic Selection | ✗ | ✗ | ✗ | ✓ | ✗ | ✓ |

*Table 2.* Different approaches for estimating ATE and the corresponding selection bias that they aim to correct. IPW uses the naive inverse propensity weighting on the selected population, completely neglecting the bias that selection poses. Polynomial estimator assumes a polynomial relation from the covariate to the outcome, which is suitable for extrapolation, i.e. tackling deterministic selection, but not adjusting for non-deterministic selection. MLE and score matching without correction are also suitable for non-deterministic selection, given the underlying distribution satisfies our smoothness condition (Proposition 3.3) and the estimators can approximate such densities well. Finally, the MLE and score matching with correction ($+\beta$) that estimates the selection function are the proposed methods suitable for *any* selection (both deterministic and non-deterministic).

the biased density observed in the dataset. We assume a relationship governed by a selection probability function $\beta(x, y, t)$:

$$\tilde{p}(y|x) \propto p(y|x) \cdot \beta(x, y, t) \quad (3)$$

We parameterize the underlying score function as $s_\theta(x, y)$ and the selection weight as $\beta_\phi(x, y, t)$. By taking the gradient of the log-density with respect to $y$, the score of the *observed* distribution, $\psi$, decomposes additively:

$$\psi(x, y, t) \triangleq \nabla_y \log \tilde{p}(y|x)$$
$$= s_\theta(x, y) + \nabla_y \log \beta_\phi(x, y, t) \quad (4)$$

Crucially, the intractable partition function of the density depends only on $x$ and $t$, and thus vanishes under $\nabla_y$.

We jointly optimize $\theta$ and $\phi$ by minimizing the following objective over the observed samples:

$$\mathcal{L}(\theta, \phi) = \mathbb{E}_{\mathcal{D}_{\text{obs}}} \Big[ \underbrace{\tfrac{1}{2}\|\psi\|^2 + \text{tr}(\nabla_y \psi)}_{\text{Hyvärinen Score Matching}}$$
$$- \lambda_1 \underbrace{\log \beta_\phi}_{\text{Likelihood}} + \lambda_2 \underbrace{\|\beta_\phi\|^2}_{\text{Regularization}} \Big] \quad (5)$$

where $\psi$ is evaluated at $(x, y, t)$. The first term ensures the composite model fits the observed data structure. The second term ($-\log \beta_\phi$) maximizes the likelihood that the observed samples were selected (i.e., $\beta \to 1$ for observed data). The third term regularizes the selection weights.

## 5. Experiments

In this section, we evaluate the performance of our proposed methods on both synthetic and real-world datasets. In the synthetic setting (Section 5.1), we focus on estimating the ATE under various noise distributions and function forms[2]. Then in Section 5.2, we perform semi-synthetic experiments on one of the largest biomedical data resources All of Us[3] (AoS). The implementation details and additional experimental results can be found in Appendix D.

---
[2]Code implementation on the synthetic dataset: https://github.com/EvieQ01/causal_effect_id_selection_bias

[3]https://www.researchallofus.org/

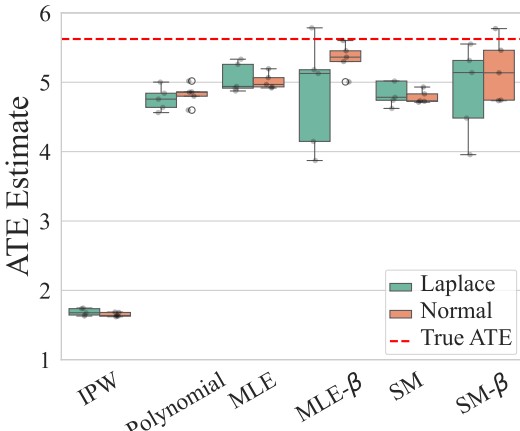

*(a)* Result of ATE: additive Gaussian/Laplace noise

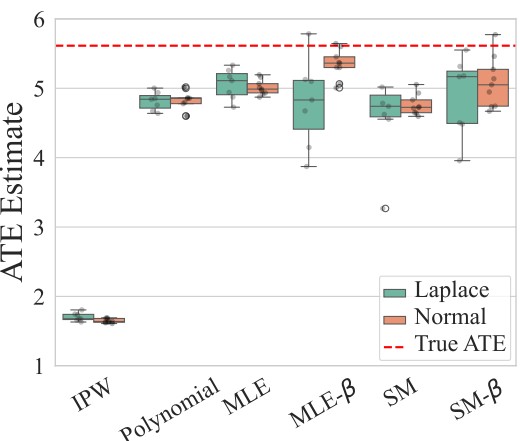

*(b)* Result of ATE: multiplicative Gaussian/Laplace noise

*Figure 2.* Comparison of ATE Estimation under different noise distributions and function types, when both deterministic and non-deterministic selection are applied. We present results for additive Gaussian and Laplace noise, and extend beyond additive noise to multiplicative noise. Overall, we observe that vanilla application of IPW leads to significantly biased estimates, while our approaches significantly improve, regardless of the specific estimator (in particular, the agnostic methods MLE+$\beta$ and SM+$\beta$).

## 5.1. Synthetic Data

The simulation data is generated using the structural equation model: $Y = f(T, X, E)$, while we draw $X$ from Uniform distribution and $E$ from different noise distributions. We consider both additive noise (i.e., $Y = f(T, X) + E$) and multiplicative noise (i.e., $Y = (1 + E) \cdot f(T, X)$) settings. The treatment variable $T$ is generated as $T = g(X)$. We then apply both deterministic and non-deterministic selection mechanism. The specific functional forms of $f, g$ and selection mechanisms are detailed in Appendix D.

**Baseline Methods** We compare our proposed method with two baselines: Naive inverse propensity weighting (IPW) and polynomial regression. The Naive IPW method estimates the ATE only using the observed part, neglecting the bias exhibited in the data. The polynomial method fits a polynomial regression model $\hat{f}_t(x) = \mathbb{E}(Y(t) \mid X = x)$ to adjust for covariates.

**Ablation** We denote the final corrected version of our proposed methods as MLE+$\beta$ and SM+$\beta$. In addition, we also include the results from the proposed methods MLE (Section 4.1) and SM (Section 4.2) without applying the selection bias adjustment step, denoted as MLE and SM. Without the selection bias adjustment, these methods can estimate the ATE more accurately than the baselines, but still suffer from bias due to the nondeterministic selection mechanism. In Table 2, we present different ATE estimator and whether they incorporate selection bias adjustment for deterministic and non-deterministic selection. We refer to Appendix E for an ablation study on different functional forms and varying selection strength.

**Results** For each run, we generate a dataset of size $N = 5000$ and apply deterministic and non-deterministic selection. After selection, there is approximately 3000 samples. We repeat the experiment 5 times and use boxplots to summarize the results (Figure 2). We demonstrate that our proposed methods (MLE+$\beta$ and SM+$\beta$) consistently outperform the baselines (IPW and Polynomial regression) across a combination of noise types (Gaussian and Laplace) and noise function forms (additive and multiplicative).

When comparing MLE+$\beta$ to its uncorrected counterpart MLE (similarly for SM+$\beta$ and SM), we observe that the +$\beta$ versions generally performs better in reducing the error but suffer from bigger variance. When comparing MLE+$\beta$ and SM+$\beta$, we observe that SM+$\beta$ generally performs slightly better in terms of lower variances, but not by a big margin. Therefore the choice between them can be subjective.

## 5.2. Semi-synthetic Data

We evaluate our proposed methods on semi-synthetic data from the All of Us Research Hub, a data-sharing platform for the National Institutes of Health (NIH) All of Us Research

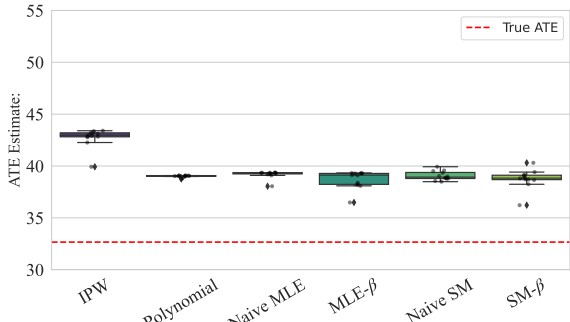

*Figure 3.* ATE estimation results on All of Us dataset. Our approach significantly decreases the bias, but not entirely. We suspect this is due to the complexity of this real-world distribution, e.g., low propensity scores ($\sim 0.05$), which makes the overlap very weak and leads to challenging estimation.

Program. It is a longitudinal cohort study aimed at enrolling over one million participants to build one of the most diverse health databases in history, specifically targeting populations historically underrepresented in biomedical research.

Among the variables, we retrieve Type 2 diabetes (T2D) as the exposure, and BMI as the covariate. Since we do not have the true causal graph between the variables, and the wide presence of genetics (Pingault et al., 2021; Zhao et al., 2024) as hidden confounders, we generate a synthetic outcome $Y$ based on the structural equation model $Y = f(T, X) + E$. We argue that when the data is composed in this manner, we have the distribution over covariate and exposure that reflects the real-world scenario, while we have the ground truth of ATE since we generate the outcome and we can make sure that the conditions proposed in Proposition 3.3 and Proposition 3.5 are met. The outcome could be imagined as Hemoglobin A1c (HbA1c) level, but note that we do not intend to draw any conclusions about the ATE from the synthetic data for any public health advice purposes.

In particular, we retrieve the last recorded BMI and T2D diagnosis status from each individual. We consider the value of BMI that is between 10 and 80 (exclude individuals with extreme BMI values to avoid outliers, the full support of BMI is $[0, 100]$). We then generate the outcome $Y$ using $N = 10000$ samples from the All of Us dataset. We consider the additive Gaussian noise setting, since the underlying distribution is already challenging. The functional form of $f$ and the selection mechanism are detailed in Appendix D. After applying a non-deterministic selection mechanism, there are approximately 2500 samples left. We repeat the experiment 5 times and summarize the results in Figure 3. Similar to the synthetic experiments, our proposed methods (MLE+$\beta$ and SM+$\beta$) consistently outperform the baselines. While the bias is not fully corrected, we remark that this experiment is based on the distributions of real-world data, which can be arbitrarily complex, making

the estimation particularly difficult. Nevertheless, our method substantially decreases the bias, making it attractive even in these challenging settings as no alternative exists.

## 6. Conclusion

In this work, we have explored the identifiability of ATE under general selection bias, moving beyond traditional graphical criteria to a more holistic, distribution-based understanding. We established necessary and sufficient conditions for s-ATE-full identifiability (Condition 1), and provide instantiation of distributions that satisfy such conditions. From there, we characterized the classes of propensity scores, covariate-outcome distributions, and selection probabilities required for recovery. Different than prior work, our solution leverages mild distributional assumptions that are independent of the specific graphical structure and on which variables the selection happens. This enables us to give identifiability results for settings that otherwise would not be identifiable and where no other solution currently exists (Zhang et al., 2016; Correa & Bareinboim, 2017; Correa et al., 2019). At the same time, our Condition 1 is necessary and independent of specific debiasing approaches, and we have shown in Section 3.3 that existing identification results naturally follow it.

**Limitation and future work.** The limitation of our work lies in the assumptions made regarding the distribution classes that we need in order to guarantee s-ATE-full identifiability. Even though these assumption are already weaker than previous works (Correa & Bareinboim, 2017; Correa et al., 2019), future work could still consider relaxing these assumptions. It would also be interesting to employ this framework to causal discovery under selection bias, which is only partially answered in previous works (Spirtes et al., 2001; Zhang et al., 2016), and consider also to take advantage of the findings for experimental design(He & Geng, 2008; Hyttinen et al., 2013; Kocaoglu et al., 2017; Ghassami et al., 2018), e.g. the strategy for active sampling given a limited budget while still guaranteeing s-ATE-full identifiability. Further, we think there is room for improvement in the estimation techniques, which may fall short on complex real-world data distributions. Our paper laid the groundwork for methodological advances in this direction, having defined the sufficient conditions for identifiability and provided initial estimators.

## Impact Statement

This paper presents work whose goal is to advance the field of Machine Learning. There are many potential societal consequences of our work, as selection bias is an important problem in causal inference and epidemiology, so our theory can have a vast range of positive applications. We do not find any risk that we feel must be specifically highlighted here.

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

## A. Related Works

**Broader Context: Censored Data, Missing Data**   Inspired by a reviewer, we clarify the scope of the problem in this paper in relation to the broader literature of *censored data* and *missing data* problem. Heckman's well-known *censoring* (Heckman, 1977) problem setting assumes that certain covariates are observed for all individuals, including non-selected ones, which we do *not* assume to have in our setting. In contrast, our setting falls within the *truncation* regime in Heckman's terminology: non-selected units are entirely absent from the dataset, and we observe only $(X_i, Y_i, T_i)$ for individual $i$ with $S_i = 1$. This applies to both our deterministic selection (where a region is completely unobserved) and nondeterministic selection (where data points are probabilistically excluded). Censored data is also commonly referred to as missing data in the modern causal inference and statistics literature. This connection was formalized by Rubin's missing data framework (Little & Rubin, 2019), which classifies missingness mechanisms into three categories: missing completely at random (MCAR), missing at random (MAR), and missing not at random (MNAR). Under the MAR assumption, where the probability of missingness depends only on observed variables is the logic underlying Heckman's (Heckman, 1977) *censoring* paradigm.

Note that, in the paper, we distinguish a deterministic (also stated as hard truncation, Proposition 3.3) and non-deterministic selection mechanism (soft selection, Proposition 3.5). The hard truncation here refers to a specific kind of truncation in the Heckman's sense: the cases where the entire region of the space is unobserved, while the soft selection refers to another kind of "truncation": the cases when every point in the support still has a positive probability of being observed.

**Causal Inference under selection bias**   For the purpose of causal inference (bias adjustment), the nonparametric perspective builds on the graphical representation of selection mechanisms, as introduced in the selection diagram framework (Bareinboim et al., 2014; Bareinboim & Pearl, 2012; Bareinboim & Tian, 2015; Correa et al., 2019; Bareinboim et al., 2022), which characterizes conditions under which causal effects remain identifiable despite selection bias. Subsequent works extended this approach by developing testable implications of selection mechanisms (Correa et al., 2019) and providing adjustment criteria. Additionally, (Van der Zander et al., 2014; Jaber et al., 2019) developed causal effect identification results under Markov equivalence classes (MEC), which take into account the potential presence of latent selections. Selection bias has been studied extensively in economics (Heckman, 1977) and epidemiology (Robins et al., 2000). Selection bias has been studied extensively in economics (Heckman, 1977) and epidemiology (Robins et al., 2000). Our work is distinct as it makes weak distributional assumptions that enable identifiability, building on the machinery of Taylor approximation results in statistical learning theory (Daskalakis et al., 2021).

**Causal Discovery under selection bias**   For the purpose of causal discovery, the foundational work of the FCI algorithm (Spirtes et al., 2001; Zhang, 2008) aims to discover ancestral relations up to an equivalence class in the presence of latent confounders and selection bias. A line of work using nonparametric methods has been developed to identify causal relations based on conditional independence constraints (Hernán et al., 2004; Tillman & Spirtes, 2011; Evans & Didelez, 2015; Versteeg et al., 2022a). There are recent works that deal with selection bias in interventional studies (Dai et al., 2025a) and in sequential data (Zheng et al., 2024; Qiu et al., 2024), and with rank constraint (Dai et al., 2025b). Several other approaches have been developed for local causal orientation (Zhang et al., 2016; Versteeg et al., 2022b), and applications in gene regulatory network inference (Luo et al., 2025). While we do not look at the causal discovery setting, we believe that a similar definition of identifiability could be formulated.

## B. Definitions and Proofs

### B.1. More Definitions

**Definition B.1** (Compatibility). A tuple $(P_{t|x}, P_{xy(t)}, P_{s|xyt})$ is said to be compatible with distribution classes $(\mathbb{P}_{t|x}, \mathbb{P}_{xy(t)}, \mathbb{S})$ if $P_{t|x} \in \mathbb{P}_{t|x}, P_{xy(t)} \in \mathbb{P}_{xy(t)}, P_{s|xyt} \in \mathbb{S}$ and $(P_{t|x}, P_{xy(t)})$ defines a valid distribution over $(X, T, Y(0), Y(1))$.

**Definition B.2** (Realizability). A (underlying) distribution $\mathcal{P}(\mathbf{V})$ is said to be realizable with respect to the distribution class pair $(\mathbb{P}_{t|x}, \mathbb{P}_{t|xy(t)})$ if the propensity score $P_{t|x}$ induced by $\mathcal{P}(\mathbf{V})$ belong to $\mathbb{P}_{t|x}$ and $\mathcal{P}_{X,Y(t)} \in \mathbb{P}_{t|xy(t)}$ denoted as $\mathcal{P}(\mathbf{V}) \sim (\mathbb{P}_{t|x}, \mathbb{P}_{xy(t)})$. After the selection procedure is applied to $\mathcal{P}(\mathbf{V})$ by any $P_{s|xyt}$ that belongs to a distribution class $\mathbb{S}$, the resulting observational study $\mathcal{P}(\mathbf{V} \mid S = 1)$ is said to be realizable with respect to $(\mathbb{P}_{t|x}, \mathbb{P}_{xy(t)}, \mathbb{S})$.

**Definition B.3** (ATE identifiablity). Intuitively, if two observational studies satisfy $\mathcal{P}^1(\mathbf{V}) = \mathcal{P}^2(\mathbf{V})$, then it should be $\tau_{\mathcal{P}^1} = \tau_{\mathcal{P}^2}$. Formally, we say that an ATE $\tau_{\mathcal{P}}$ is *identifiable* from the distribution $\mathcal{P}(\mathbf{V}) \sim (\mathbb{P}_{t|x}, \mathbb{P}_{xy(t)})$ if there is a mapping $f$ such that $f(\mathcal{P}(\mathbf{V})) = \tau_{\mathcal{P}}$ for any observational study $\mathcal{P}$ realizable with respect to such $(\mathbb{P}_{t|x}, \mathbb{P}_{xy(t)})$. (The distribution class pair $(\mathbb{P}_{t|x}, \mathbb{P}_{xy(t)})$ needs to satisfy *some* condition in order for ATE to be identifiable.)

**Assumption B.4** (Standard assumptions for ATE identifiability)**.** The following standard assumptions are made throughout the literature to ensure the identifiability of ATE (Rosenbaum & Rubin, 1983; Imbens & Rubin, 2015):

- Unconfoundedness (Ignorability):. Conditional on observed covariates ($X$), treatment assignment is independent of the potential outcomes, i.e. $Y(0), Y(1) \perp\!\!\!\perp T \mid X$

- Positivity (Overlap): Every individual in every subgroup has a non-zero probability of being in either the treatment or control group, i.e. $(0 < P(T = 1 \mid X) < 1)$.

- Consistency: The observed outcome equals the potential outcome under the treatment actually received: $Y = TY_1 + (1 - T)Y_0$

- Stable Unit Treatment Value Assumption (SUTVA): A unit's outcome is unaffected by the treatment status of other units (no interference), and there are no hidden versions of the treatment.

## B.2. ATE Identifiability for full population with external unbiased $P_X$

**Condition 2** (s-ATE-full Condition with external data)**.** The distribution classes $(\mathbb{P}_{t|x}, \mathbb{P}_{xy(t)}, \mathbb{S})$ satisfy the *Identifiability Condition under Selection* if for any tuples $(P_{t|x}, P_{xy(t)}, P_{s|xyt}), (Q_{t|x}, Q_{xy(t)}, Q_{s|xyt}) \in \mathbb{P}_{t|x} \times \mathbb{P}_{xy(t)} \times \mathbb{S}$ that are compatible with $(\mathbb{P}_{t|x}, \mathbb{P}_{xy(t)}, \mathbb{S})$, it holds that

$$\underbrace{\tau_{P_{xy(t)}} \neq \tau_{Q_{xy(t)}}}_{①} \implies \underbrace{P_X \neq Q_X}_{③}, \text{or} \quad \exists(x, y, t) \in \mathcal{A},$$

$$\text{s.t.} \quad \underbrace{\alpha_P(x, y, t)P_{t|x}P_{xy(t)} \neq \alpha_Q(x, y, t)Q_{t|x}Q_{xy(t)}}_{②},$$

where $\alpha_P := \frac{P_{s|xyt}}{P(s)}, \alpha_Q := \frac{Q_{s|xyt}}{Q(s)}$.

**Theorem B.5** (Characterization of s-ATE-full with external data)**.** *The ATE $\tau_P$ is identifiable from any observational distribution $P$ and unbiased marginal $P(X)$ realizable with respect to $(\mathbb{P}_{t|x}, \mathbb{P}_{xy(t)}, \mathbb{S})$ if and only if $(\mathbb{P}_{t|x}, \mathbb{P}_{xy(t)}, \mathbb{S})$ satisfy Condition. 2.*

The proof of Theorem B.5 follows a similar logic to the main proof (of Theorem 3.1).

## B.3. ATE Identifiability for subpopulation

When we consider the ATE for subpopulation $P(\mathbf{V} \mid S)$, we essentially consider the selected distribution as a new distribution $P'(\mathbf{V}) := P(\mathbf{V} \mid S)$. Then the ATE for subpopulation is equivalent to the ATE for the new distribution $P'$, i.e. $\tau_{P'} = \mathbb{E}_{P'}[Y(1)] - \mathbb{E}_{P'}[Y(0)]$. The following Theorem. B.6 characterize the identifiability of ATE from $P'$, and Proposition. B.7 is the instantiation. Theorem B.6 is the standard ATE identifiability result without considering selection bias (Cai et al., 2025).

**Condition 3** (s-ATE-sub Condition)**.** The distribution classes $(\mathbb{P}_{t|x}, \mathbb{P}_{xy(t)})$ satisfy the *Identifiability Condition* if for any tuples $(P_{t|x}, P_{xy(t)}), (Q_{t|x}, Q_{xy(t)}) \in \mathbb{P}_{t|x} \times \mathbb{P}_{xy(t)}$ that are compatible with $(\mathbb{P}_{t|x}, \mathbb{P}_{xy(t)})$, then:

$$\underbrace{\tau_{P_{xy(t)}} \neq \tau_{Q_{xy(t)}}}_{①} \implies \underbrace{P_X \neq Q_X}_{②} \text{ or}$$

$$\exists(x, y, t) \in \mathcal{A}, s.t., \underbrace{P_{t|x}(x)P_{xy(t)}(x, y) \neq Q_{t|x}(x)Q_{xy(t)}(x, y)}_{④}$$

**Theorem B.6** (Identification of ATE)**.** *The ATE $\tau_P$ is identifiable from any observational distribution $P$ realizable with respect to $(\mathbb{P}_{t|x}, \mathbb{P}_{xy(t)})$ if and only if $(\mathbb{P}_{t|x}, \mathbb{P}_{xy(t)})$ satisfy Condition 3.*

**Proposition B.7.** $(\mathbb{P}_{t|x}(c), \mathbb{P}_{xy(t)})$ satisfy Condition 3, $\forall c, 0 < c < \frac{1}{2}$.

**B.4. Proof of Theorems**

**Theorem 3.1** (Characterization of s-ATE-full)**.** *The ATE $\tau_P$ is identifiable from any observational distribution $P$ realizable with respect to $(\mathbb{P}_{t|x}, \mathbb{P}_{xy(t)}, \mathbb{S})$ if and only if $(\mathbb{P}_{t|x}, \mathbb{P}_{xy(t)}, \mathbb{S})$ satisfy Condition. 1.*

The following proof arguments follows in part (Cai et al., 2025, Theorem 1.1).

**Proof Sketch (Sufficiency)**  First, we show the existence of a mapping $g$ from the observed distribution $P_{\mathrm{obs}} = \mathcal{P}(\mathbf{V} \mid S = 1)$ to a function $g(P_{\mathrm{obs}}) : \mathbb{R}^d \times \mathbb{R} \times \{0, 1\} \to \mathbb{R}$ such that $g(P_{\mathrm{obs}})$ identifies a subset of candidates in $(\mathbb{P}_{t|xy(t)}, \mathbb{P}_{xy(t)}, \mathbb{S})$ that are compatible with $P_{\mathrm{obs}}$. Further, by the definition of Condition 1, for any two candidates in this subset, and any $t \in \{0, 1\}$ the expected values of $Y(t)$ under their outcome distributions are the same.

**Proof Sketch (Necessity)**  We can construct two distribution $P_{\mathrm{obs}}$ and $Q_{\mathrm{obs}}$ such that they are realizable with respect to $(\mathbb{P}_{t|xy(t)}, \mathbb{P}_{xy(t)}, \mathbb{S})$, but their ATE are different.

*Proof of Thm. 3.1.* **(Sufficiency)** We assume that the distribution classes $(\mathbb{P}_{t|x}, \mathbb{P}_{xy(t)}, \mathbb{S})$ satisfy Condition 1. Let any $\mathcal{P} = P(V)$ be an underlying distribution realizable with respect to $(\mathbb{P}_{t|x}, \mathbb{P}_{xy(t)}, \mathbb{S})$, with the corresponding observed distribution under selection $P_{\mathrm{obs}} := P(V \mid S = 1)$, where $V = (X, Y(T), T)$. Let $S_{\mathrm{obs}}$ be the set of all such observed distributions under selection, that is

$$S_{\mathrm{obs}} := \{P_{\mathrm{obs}} \mid P_{\mathrm{obs}} \text{ realizable w.r.t. } (\mathbb{P}_{t|x}, \mathbb{P}_{xy(t)}, \mathbb{S})\} \subseteq \Delta\left(\mathbb{R}^d \times \mathbb{R} \times \{0, 1\}\right).$$

We will construct a function $f : S_{\mathrm{obs}} \to \mathbb{R}$ such that $f(P_{\mathrm{obs}}) = \tau_{\mathcal{P}}$, thus proving that the ATE is identifiable.

Towards that end, let us fix one such underlying distribution $\mathcal{P}$ realizable with respect to $(\mathbb{P}_{t|x}, \mathbb{P}_{xy(t)}, \mathbb{S})$, with the corresponding observed distribution under selection $P_{\mathrm{obs}}$. We set the value of the function $f$ to be $f(P_{\mathrm{obs}}) = \tau_{\mathcal{P}} = \mathbb{E}_{\mathcal{P}}[Y(1) - Y(0)]$. It is left to prove that $f$ is well-defined, i.e., for any other underlying distribution $\mathcal{Q}$ realizable with respect to $(\mathbb{P}_{t|x}, \mathbb{P}_{xy(t)}, \mathbb{S})$, and observed distribution under selection $Q_{\mathrm{obs}} = Q(V \mid S = 1)$ it holds

$$Q_{\mathrm{obs}} = P_{\mathrm{obs}} \implies \tau_{\mathcal{Q}} = f(P_{\mathrm{obs}}).$$

By definition of realizability, it holds $Q_{t|x}, Q_{xyt} \in (\mathbb{P}_{t|x}, \mathbb{P}_{xy(t)})$. Moreover, by assumption that $(\mathbb{P}_{t|x}, \mathbb{P}_{xy(t)}, \mathbb{S})$ satisfies Condition 1, and using contraposition of the implication, it holds

$$\underbrace{\forall (x, y, t) \in \mathcal{A} \ \alpha_P(x, y, t) P_{t|x} P_{xy(t)} = \alpha_Q(x, y, t) Q_{t|x} Q_{xy(t)}}_{\neg②} \implies \underbrace{\tau_{P_{xy(t)}} = \tau_{Q_{xy(t)}}}_{\neg①},$$

where by definition $\tau_{P_{xy(t)}} = \tau_{\mathcal{P}}$, and $\tau_{Q_{xy(t)}} = \tau_{\mathcal{Q}}$.

By Bayes rule and assumed unconfoundedness, we have

$$\begin{aligned} Q_{\mathrm{obs}}(X = x, Y = y(t), T = t) &= Q(X = x, Y = y(t), T = t \mid S = 1) \\ &= \frac{Q(S = 1 \mid X = x, Y = y(t), T = t) Q(X = x, Y = y(t), T = t)}{Q(S = 1)} \\ &= \alpha_Q Q_{t|x} Q_{xy(t)}. \end{aligned}$$

Similarly $P_{\mathrm{obs}}(X = x, Y = y(t), T = t) = \alpha_P P_{t|x} P_{xy(t)}$, and from $P_{\mathrm{obs}} = Q_{\mathrm{obs}}$ follows $\neg③$. Using Condition 1 gets that $\neg①$ holds, so

$$\tau_{\mathcal{P}} = \tau_{\mathcal{Q}},$$

thus proving that the function $f$ is indeed well-defined.

$\square$

**Necessity.**  Suppose that $(\mathbb{P}_{t|x}, \mathbb{P}_{xy(t)}, \mathbb{S})$ does not satisfy Condition 1. We will show that in this case, for any map $f : \Delta(\mathbb{R}^d \times \mathbb{R} \times \{0, 1\}) \to \mathbb{R}$, there exists a distribution $P(V \mid S)$ realizable with respect to $(\mathbb{P}_{t|x}(c), \mathbb{P}_{xy(t)}, \mathbb{S})$ such that $f(P(V \mid S)) \neq \tau_P$. That is, ATE is not identifiable over this class.

By the negation of Condition 1, two tuples

$$(P_{t|x}, P_{xy(t)}, P_{s|xyt}), (Q_{t|x}, Q_{xy(t)}, Q_{s|xyt}) \in (\mathbb{P}_{t|x}, \mathbb{P}_{xy(t)}, \mathbb{S}),$$

such that ① holds, but ② does not. More precisely, it holds

$$\underbrace{\tau_{P_{xy(t)}} \neq \tau_{Q_{xy(t)}},}_{①} \quad \underbrace{\forall (x,y,t) \in \mathcal{A} \ \alpha_P(x,y,t) P_{t|x} P_{xy(t)} = \alpha_Q(x,y,t) Q_{t|x} Q_{xy(t)},}_{\neg②}$$

where $\alpha_P := \frac{P_{s|xyt}}{P(s)}, \alpha_Q := \frac{Q_{s|xyt}}{Q(s)}$. We will use these two tuples to construct a counterexample. Recall that $V$ denotes the joint random variable, i.e., $V = (X, Y, T)$. We now show that observational studies $P(V \mid S), Q(V \mid S)$ induced by $\mathcal{P}, \mathcal{Q}$, respectively, coincide.

By Bayes' rule, follows

$$P(X = x, Y = y(t), T = t \mid S = 1) = \frac{P(S = 1 \mid X = x, Y = y(t), T = t) P(X = x, Y = y(t), T = t = 0)}{P(S = 1)}.$$

Using the factorization $P(X = x, Y = y(t), T = t) = P_{t|x} P_{xy(t)}$, following from uconfoundedness, and the definition $\alpha_P = \frac{P_{s|xyt}}{P(s)} = \frac{P(S=1|X=x,Y=y(t),T=t)}{P(S=1)}$, we obtain

$$P(X = x, Y = y(t), T = t \mid S = 1) = \alpha_P P_{t|x} P_{xy(t)}.$$

By $\neg②$ this equals $\alpha_Q Q_{t|x} Q_{xy(t)}$, and hence by another use of Bayes rule, we get

$$P(X = x, Y = y(t), T = t \mid S = 1) = Q(X = x, Y = y(t), T = t \mid S = 1),$$

yielding $P(V \mid S = 1) = Q(V \mid S = 1)$. Despite having the same observational studies, the ATE will differ. In particular, as the assumed property ① holds, we have

$$\tau_{\mathcal{P}} = \mathbb{E}_{\mathcal{P}}[Y(1)] - \mathbb{E}_{\mathcal{P}}[Y(0)] \neq \mathbb{E}_{\mathcal{Q}}[Y(1)] - \mathbb{E}_{\mathcal{Q}}[Y(0)] = \tau_{\mathcal{Q}}.$$

From this follows that there cannot exist a function $f : \Delta(\mathbb{R}^d \times \mathbb{R} \times \{0,1\}) \to \mathbb{R}$, such that both $\tau_P = f(P(V \mid S)) = f(Q(V \mid S)) = \tau_Q$. Thus, ATE is not identifiable with respect to $(\mathbb{P}_{t|x}, \mathbb{P}_{xy(t)}, \mathbb{S})$, proving the claim.

### B.5. Proof of Propositions

**Proposition 3.3** (Cases under Deterministic Selection). *The tuple $(\mathbb{P}_{t|x}(c), \mathbb{P}_{xy(t)}^{C^\infty}, \mathbb{S}_{\mathbb{1}}^{\det})$ satisfies Condition 1, $\forall c, 0 < c < \frac{1}{2}$, where $\mathbb{P}_{xy(t)}^{C^\infty} \subseteq \Delta(\mathbb{R}^d \times \mathbb{R}) \times \{0,1\}$ is a family of distributions such that for any $P_{xy(t)} \in \mathbb{P}_{xy(t)}^{C^\infty}$, and for any $t \in \{0,1\}$, its corresponding conditional distribution $P_{y(t)|x}$ is parametrized as $P_{y(t)|x} \propto e^{f(x,y)}$, and its corresponding marginal distribution $P_x$ is parametrized as $P_x \propto e^{g(x)}$, where $f(x,y) = f_{P_{xy(t)}}$ and $g(x) = g_{P_{xy(t)}}$ are polynomial functions of $(x,y)$, and $x$, respectively.*

The following proof arguments follows in part (Cai et al., 2025, Lemma 4.6).

*Proof of Prop. 3.3.* We will prove this result by contradiction. Namely, suppose that the triplet $(\mathbb{P}_{t|x}(c), \mathbb{P}_{xy(t)}^{C^\infty}, \mathbb{S}_{\mathbb{1}}^{\det})$ does not satisfy Condition 1. Equivalently, there exist two tuples $(P_{t|x}, P_{xy(t)}, P_{s|xyt}), (Q_{t|x}, Q_{xy(t)}, Q_{s|xyt}) \in \mathbb{P}_{t|x}(c) \times \mathbb{P}_{xy(t)}^{C^\infty} \times \mathbb{S}_{\mathbb{1}}^{\det}$ such that ① holds, but ② does not. More precisely, it holds

$$\underbrace{\tau_{P_{xy(t)}} \neq \tau_{Q_{xy(t)}},}_{①} \quad \underbrace{\forall (x,y,t) \in \mathcal{A} \ \alpha_P(x,y,t) P_{t|x} P_{xy(t)} = \alpha_Q(x,y,t) Q_{t|x} Q_{xy(t)},}_{\neg②}$$

where $\alpha_P := \frac{P_{s|xyt}}{P(s)}, \alpha_Q := \frac{Q_{s|xyt}}{Q(s)}$.

Take any $t \in \{0,1\}$. By definition of $\mathbb{S}_{\mathbb{1}}^{\det}$ it holds that there exist sets $\mathcal{B}_P, \mathcal{B}_Q \subseteq \mathcal{A}$ such that

$$P_{s|xyt} = \mathbb{1}_{\mathcal{A} \setminus \mathcal{B}_P} \text{ and } Q_{s|xyt} = \mathbb{1}_{\mathcal{A} \setminus \mathcal{B}_Q}.$$

Let us denote by $\mathcal{A}^t, \mathcal{B}_P^t$ and $\mathcal{B}_Q^t$ restriction of these sets to fixed $t$. For notational convenience, we continue to write $t$ as a variable, although it will henceforth always represent one instantiation of it $t \in \{0,1\}$.

We first show that $B_P^t = B_Q^t$. Let $(x, y, t) \in B_P^t$. By definition of $B_P^t$ and $\mathbb{S}_1^{\text{det}}$,

$$P(s \mid x, y, t) = 0, \text{ from which follows } \alpha_P(x, y, t) P_{t|x} P_{xy(t)} = 0.$$

Then by the assumed $\neg\text{\textcircled{3}}$ this implies $\alpha_Q(x, y, t) Q_{t|x} Q_{xy(t)} = 0$. Since $Q_{t|x} \in \mathbb{P}_{t|x}(c)$, we have $Q_{t|x} > c > 0$. Moreover, because $Q_{xy(t)} \in \mathbb{P}_{xy(t)}^{C^\infty}$,

$$Q_{xy(t)} = Q_{y(t)|x} Q_X \propto e^{f_Q(x,y(t))} e^{g_Q(x)} > 0,$$

where the strict inequality follows from $e^{f_Q(x,y(t))} e^{g_Q(x)} > 0$. Therefore, it can only be $\alpha_Q(x, y, t) = 0$, from which directly follows $Q(s \mid x, y, t) = 0$. By definition of $\mathbb{S}_1^{\text{det}}$ this yields $(x, y, t) \in B_Q^t$. Hence,

$$B_P^t \subseteq B_Q^t$$

By symmetry, the reverse inclusion also holds, and thus $B_P^t = B_Q^t$. From this, follows that

$$P(s) = \iiint_{\mathcal{A}^t \setminus \mathcal{B}_P^t} P(s \mid x, y, t) = \iiint_{\mathcal{A}^t \setminus \mathcal{B}_Q^t} Q(s \mid x, y, t) = Q(s).$$

We now restrict attention to $(x, y, t) \in \mathcal{A}^t \setminus \mathcal{B}_P^t = \mathcal{A}^t \setminus \mathcal{B}_Q^t$. On this region, by definition of $\mathbb{S}_1^{\text{det}}$,

$$P_{s|xyt} = Q_{s|xyt} = 1.$$

Combining this with $P(s) = Q(s)$, the assumption $\neg\text{\textcircled{3}}$ implies

$$P_{t|x} P_{xy(t)} = Q_{t|x} Q_{xy(t)},$$

for $(x, y, t) \in \mathcal{A}^t \setminus \mathcal{B}_P^t$. Writing this in terms of conditional distributions yields

$$P_{t|x} P_{y(t)|x} P_X = Q_{t|x} Q_{y(t)|x} Q_X. \tag{B.1}$$

Denote $C_{x,t} := (Q_{t|x} Q_X)/(P_{t|x} P_X)$, which is well defined since $Q_{t|x} \in \mathbb{P}_{t|x}(c)$, i.e., $Q_{t|x} > 0$, and $P_X > 0$ as assumed. Then for all $(x, y, t) \in \mathcal{A}^t \setminus \mathcal{B}_P^t$,

$$P_{y(t)|x} = C_{x,t} Q_{y(t)|x}.$$

Thus, on the selected sliced line $\Omega_{x,t} = \{y : (x, y, t) \in \mathcal{A}^t \setminus \mathcal{B}^t\}$, the two conditional densities are proportional by a constant that does not depend on $y$. Furthermore, by definition of $\mathbb{P}_{xy(t)}^{C^\infty}$ there exist functions $Z_P(x), Z_Q(x) > 0$ such that

$$P_{y(t)|x} = \frac{e^{f_P(x,y)}}{Z_P(x)}, \qquad Q_{y(t)|x} = \frac{e^{f_Q(x,y)}}{Z_Q(x)},$$

where $f_P(x, y)$ and $f_Q(x, y)$ are polynomials in $(x, y)$ and $Z_P(x)$ and $Z_Q(x)$ are normalizing constants such that $Z_P(x) = \int e^{f_P(x,y)} dy$, $Z_Q(x) = \int e^{f_Q(x,y)} dy$. Substituting into (B.1) yields

$$e^{f_P(x,y)} = \frac{Z_P(x)}{Z_Q(x)} C_{x,t} e^{f_Q(x,y)},$$

Taking logarithm gives and subtracting gives, for all $y \in \Omega_{x,t}$,

$$f_P(x, y) - f_Q(x, y) = c_{x,t}, \qquad c_{x,t} = \log \frac{Z_P(x) C_{x,t}}{Z_Q(x)}.$$

Note that for any fixed $x$ the left hand side is polynomial in $y$, while the right hand side of this equation is a constant in $y$. Thus, by fundamental theorem of algebra, this equation can have finite number of solutions in $\mathbb{R}$. However, by assumption that $\mu(\mathcal{B}_P^{1\,c}) > 0$, there exists an uncountable number of points $y \in \Omega_{t,x}$ for any fixed $t, x$. This is due to the fact that any 1-dimensional projection of a measurable set of positive Lebesgue measure has positive Lebesgue measure in the projected space, and is hence uncountable. Thus, for any $y \in \mathbb{R}$

$$f_P(x, y) - f_Q(x, y) = c_{x,t}.$$

Consequently, for all $y \in \mathbb{R}$.

$$P_{y(t)|x} = \frac{e^{f_P(x,y)}}{\int e^{f_P(x,y)}dy} = \frac{e^{f_Q(x,y)+c_{x,t}}}{\int e^{f_P(x,y)+c_{x,t}}dy} = \frac{e^{f_Q(x,y)}}{\int e^{f_P(x,y)}dy} = Q_{y(t)|x}. \tag{B.2}$$

Recall, by the assumption of the proposition $P_X \propto e^{g_P(x)}$ and $Q_X \propto e^{g_Q(x)}$, so by analogous argumentation we can get

$$P_X = Q_X.$$

Multiplying both sides of equation B.2 by $P_X = Q_X$ yields

$$P_{xy(t)} = P_{y(t)|x}P_X = Q_{y(t)|x}Q_X = Q_{xy(t)},$$

for all $(x, y(t)) \in \mathbb{R}^d \times \mathbb{R}$. Therefore,

$$\mathbb{E}_{(x,y)\sim P_{xy(t)}}[y] = \mathbb{E}_{(x,y)\sim Q_{xy(t)}}[y].$$

As $t \in \{0, 1\}$ was chosen arbitrarily, we have

$$\tau_{P_{xy(t)}} = \mathbb{E}_{(x,y)\sim P_{xy(1)}}[y] - \mathbb{E}_{(x,y)\sim P_{xy(0)}}[y] = \mathbb{E}_{(x,y)\sim Q_{xy(1)}}[y] - \mathbb{E}_{(x,y)\sim Q_{xy(0)}}[y] = \tau_{Q_{xy(t)}}$$

contradicting ① and thus proving the claim.

$\square$

**Proposition 3.5** (Cases under Nondeterministic Selection)**.** *The tuple* $(\mathbb{P}_{t|x}(c), \mathbb{P}_{xy(t)}, \mathbb{S}(d))$ *satisfies Condition 1,* $\forall c, 0 < c < \frac{1}{2}$, *if* $\mathbb{P}_{xy(t)}$ *belongs to the following distribution classes: (1) Gaussian distribution, (2) Laplace distribution (light tailed distributions), and (3) Pareto distribution, (4) Log-normal distributions (heavy tailed distributions).*

*Proof of Prop. 3.5.* We define the $\mathbb{P}_{xy(t)} \subseteq \Delta(\mathbb{R}^d \times \mathbb{R}) \times \{0, 1\}$ as a family of distributions such that for any $P_{xy(t)} \in \mathbb{P}^{C^\infty}_{xy(t)}$, and for any $t \in \{0, 1\}$, its corresponding conditional distribution $P_{y(t)|x}$ is parametrized as stated in the proposition, and its corresponding marginal distribution $P_x$ is fixed. Let us consider any two different distributions $P, Q$ such that their corresponding conditional distributions satisfy

$$\underbrace{(P_{t|x}, P_{xy(t)}, P_{s|xyt})}_{P}, \underbrace{(Q_{t|x}, Q_{xy(t)}, Q_{s|xyt})}_{Q} \in (\mathbb{P}_{t|x}(c), \mathbb{P}_{xy(t)}, \mathbb{S}(d)).$$

We will prove that if ① from Condition 1 is satisfied by $P$ and $Q$, then ② holds must hold. Rewriting ②, we have that the ratio $\frac{Q(S=1)P_{s|xyt}P_{t|x}P_{xy(t)}}{P(S=1)Q_{s|xyt}Q_{t|x}Q_{xy(t)}} \neq 1$ for some $(x, y, t)$. Let us denote by $r := \frac{P(S=1)}{Q(S=1)}$ and $h(x) := \frac{Q_X}{P_X}$. Note that by definition of $\mathbb{S}(d)$, and restriction to the support of $X$, we have $r, h(x) \in (0, +\infty)$. Then ② holds when

$$\frac{P_{s|xyt}P_{t|x}P_{xy(t)}}{Q_{s|xyt}Q_{t|x}Q_{xy(t)}} \neq r \Rightarrow \frac{P_{t|x}P_{xy(t)}}{Q_{t|x}Q_{xy(t)}} \notin \left[rd, \frac{r}{d}\right]$$

$$\iff \frac{P_{xy(t)}}{Q_{xy(t)}} \notin \left[r\frac{cd}{1-c}, r\frac{1-c}{cd}\right]. \tag{B.3}$$

$$\iff \frac{P_{y|x}}{Q_{y|x}} \notin \left[h(x)r\frac{cd}{1-c}, h(x)r\frac{1-c}{cd}\right] \subset (0, +\infty).$$

The first equivalence holds because of the c-overlap assumption on the propensity scores $P_{t|x}, Q_{t|x}$, i.e. $c < P_{t|x}, Q_{t|x} < 1 - c$. And the second equivalence holds because of the assumption on the selection probabilities $d < P_{s|xyt}, Q_{s|xyt}$ in $\mathbb{S}(d)$. So it would be enough to show that for any two outcome distributions $P$ and $Q$, and for some $x$, the ratio of their conditional densities $\frac{P_{y|x}}{Q_{y|x}}$ is unbounded, thus falls out of the ratio bound $\left[h(x)r\frac{cd}{1-c}, h(x)r\frac{1-c}{cd}\right]$.

Next we prove separately for different distribution families.

1. **Gaussian distribution.** As ① is assumed to hold, there must exist some $x$, for which the conditional outcome distributions is:

$$P(y|x) = \frac{1}{\sqrt{2\pi\sigma^2}} \exp\left(-\frac{(y-\mu_P)^2}{2\sigma^2}\right),$$

$$Q(y|x) = \frac{1}{\sqrt{2\pi\sigma^2}} \exp\left(-\frac{(y-\mu_Q)^2}{2\sigma^2}\right),$$

where $\mu_P \neq \mu_Q$. Recall, this holds due to the assumption $\mathbb{P}_{xy(t)}$, and the fact that it has fixed marginal $P_X$.

Consider the ratio $R(y) = \frac{P(y|x)}{Q(y|x)}$:

$$R(y) = \frac{\frac{1}{\sqrt{2\pi\sigma^2}} \exp\left(-\frac{(y-\mu_P)^2}{2\sigma^2}\right)}{\frac{1}{\sqrt{2\pi\sigma^2}} \exp\left(-\frac{(y-\mu_Q)^2}{2\sigma^2}\right)}$$

$$= \exp\left(\frac{(y-\mu_Q)^2 - (y-\mu_P)^2}{2\sigma^2}\right)$$

Expand the squares in the numerator:

$$(y-\mu_Q)^2 - (y-\mu_P)^2 = (y^2 - 2y\mu_Q + \mu_Q^2) - (y^2 - 2y\mu_P + \mu_P^2)$$

$$= 2y(\mu_P - \mu_Q) + (\mu_Q^2 - \mu_P^2)$$

Substituting this back into the ratio:

$$R(y) = \exp\left(\frac{2(\mu_P - \mu_Q)}{\sigma^2}y + \frac{\mu_Q^2 - \mu_P^2}{2\sigma^2}\right)$$

The exponent is a linear function of $y$ of the form $Ay + B$, where $A = \frac{2(\mu_P - \mu_Q)}{\sigma^2} \neq 0$.

- If $\mu_P > \mu_Q$ (slope $A > 0$): $\lim_{y\to\infty} R(y) = \infty$ and $\lim_{y\to-\infty} R(y) = 0$.
- If $\mu_P < \mu_Q$ (slope $A < 0$): $\lim_{y\to\infty} R(y) = 0$ and $\lim_{y\to-\infty} R(y) = \infty$.

Since the range of the ratio function $R(y)$ is $(0, \infty)$, for any constant $c, d \in (0, 1/2)$, there exists a $y$ such that:

$$\frac{P(y|x)}{Q(y|x)} \notin \left[h(x)r\frac{cd}{1-c}, h(x)r\frac{1-c}{cd}\right]$$

Thus, the Gaussian family satisfies Condition 1.

2. **Laplace distribution.** Let the conditional outcome distributions be Laplace:

$$P(y|x) = \frac{1}{2b_P} \exp\left(-\frac{|y-\mu_P|}{b_P}\right)$$

$$Q(y|x) = \frac{1}{2b_Q} \exp\left(-\frac{|y-\mu_Q|}{b_Q}\right)$$

Consider the ratio $R(y) = \frac{P(y|x)}{Q(y|x)}$:

$$R(y) = \frac{\frac{1}{2b_P} \exp\left(-\frac{|y-\mu_P|}{b_P}\right)}{\frac{1}{2b_Q} \exp\left(-\frac{|y-\mu_Q|}{b_Q}\right)}$$

$$= \frac{b_Q}{b_P} \exp\left(\frac{|y-\mu_Q|}{b_Q} - \frac{|y-\mu_P|}{b_P}\right)$$

For sufficiently large $y$ (specifically $y > \max(\mu_P, \mu_Q)$), we have $|y - \mu| = y - \mu$. The exponent becomes:

$$E(y) = \frac{y - \mu_Q}{b_Q} - \frac{y - \mu_P}{b_P}$$

$$= y\left(\frac{1}{b_Q} - \frac{1}{b_P}\right) - \left(\frac{\mu_Q}{b_Q} - \frac{\mu_P}{b_P}\right)$$

This is a linear function of $y$ with slope $A = \frac{1}{b_Q} - \frac{1}{b_P}$.

Due to ① for some $x$ it must be $b_P \neq b_Q$, the slope $A \neq 0$.

- If $b_P > b_Q$ (slope $A > 0$), then $\lim_{y \to \infty} R(y) = \infty$.
- If $b_P < b_Q$ (slope $A < 0$), then $\lim_{y \to \infty} R(y) = 0$.

In either case, the ratio falls out of the bounded interval $\left[h(x)r\frac{cd}{1-c}, h(x)r\frac{1-c}{cd}\right]$, satisfying Condition 1.

3. **Pareto distribution.** Let the distributions be Pareto (Type I) with scale $y_m$ and shapes $\alpha_P, \alpha_Q$:

$$P(y|x) = \frac{\alpha_P y_m^{\alpha_P}}{y^{\alpha_P + 1}} \quad \text{for } y \geq y_m$$

$$Q(y|x) = \frac{\alpha_Q y_m^{\alpha_Q}}{y^{\alpha_Q + 1}} \quad \text{for } y \geq y_m$$

Consider the ratio $R(y)$ for $y \geq y_m$:

$$R(y) = \frac{\alpha_P y_m^{\alpha_P} y^{-(\alpha_P + 1)}}{\alpha_Q y_m^{\alpha_Q} y^{-(\alpha_Q + 1)}}$$

$$= \left(\frac{\alpha_P}{\alpha_Q} y_m^{\alpha_P - \alpha_Q}\right) \frac{y^{\alpha_Q + 1}}{y^{\alpha_P + 1}}$$

$$= C \cdot y^{\alpha_Q - \alpha_P}$$

where $C$ is a constant independent of $y$.

The behavior depends on the difference in shape parameters:

- If $\alpha_Q > \alpha_P$: The exponent is positive, so $\lim_{y \to \infty} R(y) = \infty$.
- If $\alpha_Q < \alpha_P$: The exponent is negative, so $\lim_{y \to \infty} R(y) = 0$.

Since there exists $x$ such that $\alpha_P \neq \alpha_Q$, due to assumed ①, the density ratio takes values in $(0, \infty)$ (or effectively unbounded intervals), ensuring it falls outside any finite overlap bounds $\left[h(x)r\frac{cd}{1-c}, h(x)r\frac{1-c}{cd}\right]$. Thus, Condition 1 is satisfied.

4. **Log-normal distribution.** Let $Y \sim \text{Lognormal}(\mu, \sigma^2)$. The density is:

$$f(y) = \frac{1}{y\sigma\sqrt{2\pi}} \exp\left(-\frac{(\ln y - \mu)^2}{2\sigma^2}\right)$$

For a fixed $x$ we denote appropriately $\mu_P$ for $P$, and $\mu_Q$ for $Q$.

Consider the ratio $R(y)$ for $y > 0$:

$$R(y) = \frac{\frac{1}{y\sigma\sqrt{2\pi}} \exp\left(-\frac{(\ln y - \mu_P)^2}{2\sigma^2}\right)}{\frac{1}{y\sigma\sqrt{2\pi}} \exp\left(-\frac{(\ln y - \mu_Q)^2}{2\sigma^2}\right)}$$

$$= \exp\left(\frac{(\ln y - \mu_Q)^2 - (\ln y - \mu_P)^2}{2\sigma^2}\right)$$

Let $z = \ln y$. The numerator in the exponent is:

$$(z - \mu_Q)^2 - (z - \mu_P)^2 = (z^2 - 2z\mu_Q + \mu_Q^2) - (z^2 - 2z\mu_P + \mu_P^2)$$
$$= 2z(\mu_P - \mu_Q) + (\mu_Q^2 - \mu_P^2)$$

Substituting back $z = \ln y$:

$$R(y) = \exp\left(\frac{2(\mu_P - \mu_Q)\ln y + C}{2\sigma^2}\right) = \exp(k \ln y + C') = C'' y^k$$

where $k = \frac{\mu_P - \mu_Q}{\sigma^2}$.

Since, due to ①, there exists an $x$ for which $\mu_P \neq \mu_Q$, we have $k \neq 0$.

- If $\mu_P > \mu_Q$ ($k > 0$), then $\lim_{y \to \infty} R(y) = \infty$.
- If $\mu_P < \mu_Q$ ($k < 0$), then $\lim_{y \to \infty} R(y) = 0$.

The ratio $R(y)$ approaches 0 or $\infty$ as $y \to \infty$. Therefore, it must eventually fall outside the overlap interval $\left[h(x)r\frac{cd}{1-c}, h(x)r\frac{1-c}{cd}\right]$, satisfying Condition 1.

$\square$

## B.6. Proof of Corollaries

**Corollary 3.9** (Selection-backdoor (Correa et al., 2019) as a special case of Condition 1). *If $S$ is a child of $T$ but not a child of $X$ nor $Y$, i.e. $T \to S$, $X \nrightarrow S$, and $Y \nrightarrow S$ in $\mathcal{G}$, then (a) $X$ satisfies the* selection-backdoor *criterion in the causal graph $\mathcal{G}$, and (b) the distribution classes $(\mathbb{P}_{t|x}, \mathbb{P}_{xy(t)})$ that are entailed by $\mathcal{G}$ satisfy Condition 1.*

*Proof of Corollary 3.9.* (a) Prove that $X$ satisfies the *selection-backdoor* criterion. If $Y \perp\!\!\!\perp S \mid T$ in $\mathcal{G}_{\overline{T}}$, then the following conditions hold: (1) $X$ is not a descendant of $T$. (2) All non-causal paths between $T$ and $Y$ in $\mathcal{G}$ are blocked by $X$ and $S$. (3) $Y \perp\!\!\!\perp S \mid T$ in $\mathcal{G}_{\overline{T}}$. (4) $T \perp\!\!\!\perp Y$ in $\mathcal{G}_{\underline{T}}$ if $T \in Anc(S)$. (b) Prove that the distribution classes $(\mathbb{P}_{t|x}, \mathbb{P}_{xy(t)})$ that are entailed by $\mathcal{G}$ satisfy Condition 1.

Take the countrapositive of Condition 1, i.e., assume that for two distributions $\mathcal{P}$ and $\mathcal{Q}$, if $\forall(x, y, t) \in \mathbb{R}^d \times \mathbb{R} \times \{0, 1\}$, $P(s \mid x, y, t) \cdot P(t \mid x) \cdot P(x, y(t)) = Q(s \mid x, y, t) \cdot Q(t \mid x) \cdot Q(x, y(t))$, then because $Y \perp\!\!\!\perp S \mid T$, so $P(y|t, s) = P(y|t)$. Hence, $P(y|t) = Q(y|t)$ and thus $\mathbb{E}_{(x,y) \sim P_{X,Y}}[y] = \mathbb{E}_{(x,y) \sim Q_{X,Y}}[y]$. Therefore, Condition 1 holds. $\square$

**Corollary 3.10** (Selection-backdoor w/ external unbiased covariate (Correa et al., 2019) as a special case of Condition 1). *If $S$ is a child of $X$ but not a child of $Y$, i.e. $X \to S$ and $Y \nrightarrow S$, then (a) $X$ satisfies the* selection-backdoor-ext *criterion in the causal graph $\mathcal{G}$, and (b) the distribution classes $(\mathbb{P}_{t|x}, \mathbb{P}_{xy(t)}, \mathbb{S})$ that are entailed by $\mathcal{G}$ satisfy Condition 1.*

*Proof of Corollary 3.10.* (a) Prove that $X$ satisfies the *selection-backdoor-ext* criterion. If $Y \perp\!\!\!\perp S \mid T, X$ in $\mathcal{G}_{\overline{T}}$, then the following conditions hold: (1) $X$ is not a descendant of $T$. (2) All non-causal paths between $T$ and $Y$ in $\mathcal{G}$ are blocked by $X$ and $S$. (3) $Y \perp\!\!\!\perp S \mid \{T, X\}$ in $\mathcal{G}$,

(b) Prove that the distribution classes $(\mathbb{P}_{t|x}, \mathbb{P}_{xy(t)}, \mathbb{S})$ that are entailed by $\mathcal{G}$ satisfy Condition 1. Take the countrapositive of Condition 1, i.e., assume that for two distributions $\mathcal{P}$ and $\mathcal{Q}$, if $\forall(x, y, t) \in \mathbb{R}^d \times \mathbb{R} \times \{0, 1\}$, $P(s \mid x, y, t) \cdot P(t \mid x) \cdot P(x, y(t)) = Q(s \mid x, y, t) \cdot Q(t \mid x) \cdot Q(x, y(t))$, then because $Y \perp\!\!\!\perp S \mid T, X$, so $P(y|t, x, s) = P(y|t, x)$. Hence, $P(y|t, x) = Q(y|t, x)$. Thus,

$$\mathbb{E}_P[y] = \int y(t)P(y)dy$$
$$= \int y \int P(y \mid x)P(x)dxdy$$
$$= \int y \int P(y \mid x, s)P(x)dxdy \qquad (B.4)$$
$$= \iint yP(y \mid x, s)P(x)dxdy$$
$$= \iint yQ(y \mid x, s)P(x)dxdy,$$

and similarly $\mathbb{E}_Q[y] = \iint yQ(y \mid x,s)Q(x)dxdy$. When $P_X(x) = Q_X(x)$ (i.e. we have extra access about unbiased covariate), we get $\mathbb{E}_P[y] = \mathbb{E}_Q[y]$. Therefore, Condition 1 holds.

$\square$

**Corollary 3.12** (Extension of outcome-dependent selection (Zhang et al., 2016))**.** *If the selection mechanism is outcome-dependent, i.e., there exist $Y \to S$ in $\mathcal{G}$, then the distribution classes $(\mathbb{P}_{t|x}, \mathbb{P}_{xy(t)}, \mathbb{S})$ that are entailed by $\mathcal{G}$ satisfy Condition 1 when they belong to the distribution classes as states in Proposition 3.3 and Proposition 3.5.*

*Proof of Corollary 3.12.* The proof directly follows from Proposition 3.3 and Proposition 3.5. $\square$

**Corollary 3.14** (S-id (Abouei et al., 2024) graphical criteria as a special case of Condition 3)**.** *If $S$ satisfies the* S-id *graphical criteria in the causal graph $\mathcal{G}$, in particular, $T \notin Anc(S)$, then the distribution classes $(\mathbb{P}_{t|x}, \mathbb{P}_{xy(t)}, \mathbb{S})$ that are entailed by $\mathcal{G}$ satisfy Condition 3.*

*Proof of Corollary 3.14.*

$$
\begin{aligned}
\mathbb{E}_P[y] &= \int y(t)P(y)dy \\
&= \int y \int P(y(t) \mid x)P(x(t))dxdy \\
&= \int y \int P(y(t) \mid x,s)P(x)dxdy \\
&= \iint yP(y(t) \mid x,s)P(x)dxdy \\
&= \iint yQ(y(t) \mid x,s)P(x)dxdy,
\end{aligned}
\tag{B.5}
$$

and similarly $\mathbb{E}_Q[y] = \iint yQ(y \mid x,s)Q(x)dxdy$. When $T \notin Anc(S)$ and $P_X(x) = Q_X(x)$, we get the second equation $P(X(t)) = P(X)$, thus $\mathbb{E}_P[y] = \mathbb{E}_Q[y]$. Therefore, Condition 3 holds. $\square$

## C. Comparison with existing graphical criteria

In the literature, several graphical criteria have been proposed to identify causal effects under selection bias, however they characterize the point-wise identifiability of $P(Y \mid do(T))$, where $do$ represent the do-operator (Pearl, 2009), rather than the distributional-level identifiability of ATE as defined in Def. 2.2.

**Definition C.1** (Causal effect identifiablity)**.** $P(Y \mid do(T))$ is identifiable from $\mathcal{P}$ if for any two distributions $\mathcal{P}_1$ and $\mathcal{P}_2$, $\mathcal{P}_1(Y \mid do(T)) = \mathcal{P}_2(Y \mid do(T))$ whenever $\mathcal{P}_1 = \mathcal{P}_2$.

*Remark* C.2 (Relation between Def. B.3 and Def. C.1)**.** Identifiability of causal functionals (e.g. ATE) can be weaker than full distributional identifiability; ATE could be identified even when the point-wise $\mathcal{P}(Y \mid do(T))$ is not. (Pearl, 2009). One can for example use the Wald estimator (with instrumental variables) (Moreira, 2003) to identify ATE under certain conditions even when $\mathcal{P}(Y \mid do(T))$ is not identifiable.

(Correa et al., 2018) Extend Pearl's backdoor adjustment to handle both confounding and selection biases when some external (unbiased) data may be available. In particular, it introduces Generalized Adjustment Criterion (GACT) for causal effect identification ($P(Y \mid X)$) when all data is biased ($P(\mathbf{V} \mid \mathbf{S} = 1)$) and when some covariates $\mathbf{Z}$ are measured without selection bias $P(\mathbf{Z})$). The two criteria are summarized below:

**Definition C.3** (GACT1)**.** $\mathbf{Z}$ satisfies the criterion w.r.t. $\mathbf{X}, \mathbf{Y}$ in $\mathcal{G}$ if:

(a) No element of $\mathbf{Z}$ is a descendant in $G_{\overline{\mathbf{X}}}$ of any $W \notin \mathbf{X}$ which lies on a proper causal path from $\mathbf{X}$ to $\mathbf{Y}$.

(b) All non-causal paths between $\mathbf{X}$ and $\mathbf{Y}$ in $G$ are blocked by $\mathbf{Z}$ and $S$.

(c) $\mathbf{Y}$ is d-separated from $S$ given $\mathbf{X}$ under the intervention, i.e., $(\mathbf{Y} \perp\!\!\!\perp S \mid \mathbf{X})_{G_{\overline{\mathbf{X}}}}$.

(d) Every $X \in \mathbf{X}$ is either a non-ancestor of $S$ or it is independent of $\mathbf{Y}$ in $G_{\underline{\mathbf{X}}}$, i.e., $\forall X \in \mathbf{X} \cap An(S)(X \perp\!\!\!\perp \mathbf{Y})_{G_{\underline{\mathbf{X}}}}$.

**Definition C.4** (GACT2)**.** $\mathbf{Z}$ satisfies the criterion w.r.t. $\mathbf{X}, \mathbf{Y}$ in $G$ if:

(a) No element of $\mathbf{Z}$ is a descendant in $G_{\overline{\mathbf{X}}}$ of any $W \notin \mathbf{X}$ which lies on a proper causal path from $\mathbf{X}$ to $\mathbf{Y}$.

(b) All non-causal paths between $\mathbf{X}$ and $\mathbf{Y}$ in $G$ are blocked by $\mathbf{Z}$.

(c) $\mathbf{Y}$ is d-separated from the selection mechanism $S$ given $\mathbf{Z}$ and $\mathbf{X}$, i.e., $(\mathbf{Y} \perp\!\!\!\perp S \mid \mathbf{X}, \mathbf{Z})$.

Then, $\mathbf{Z}$ satisfying GACT1 is neccessary and sufficient for identifying $P(Y \mid do(T))$ from biased data $P(\mathbf{V} \mid \mathbf{S} = 1)$ and unbiased data $P(\mathbf{Z})$ as $P(\mathbf{Y} \mid do(\mathbf{X})) = \sum_{\mathbf{Z}} P(\mathbf{Y} \mid \mathbf{X}, \mathbf{Z}, S = 1)P(\mathbf{Z} \mid S = 1)$. Or if external unbiased data on $\mathbf{Z}$ is available ($P(\mathbf{Z})$), $\mathbf{Z}$ satisfying GACT2 is neccessary and sufficient for identifying $P(Y \mid do(T))$ from biased data $P(\mathbf{V} \mid \mathbf{S} = 1)$ as $P(\mathbf{Y} \mid do(\mathbf{X})) = \sum_{\mathbf{Z}} P(\mathbf{Y} \mid \mathbf{X}, \mathbf{Z}, S = 1)P(\mathbf{Z})$.

However, as presented in the main paper, we argue that without a distributional-level characterization, graphical criteria such as GACT1 and GACT2 may not be sufficient to fully capture the identifiability of causal effects under selection bias. We provide cases in Table. 3 where GACT1 and GACT2 fail to identify ATE through backdoor adjustment, yet distributional conditions can still guarantee identifiability.

*Discussion* C.5 (Advantages and Disadvantages). Both frameworks have their own strengths and weaknesses. Graphical criteria (e.g., GACT1 and GACT2) are intuitive and can apply to complex causal relations. However, the disadvantages lie in that it may fail to identify causal effects in certain scenarios (e.g., output-dependent selection bias as shown in Corollary. 3.12). Further, they require the causal graph to be fully known, which may not be feasible in practice. In contrast, our framework relies on distributional assumptions that can be empirically tested.

## D. Experiment Details

In this section, we provide additional details about the experiments conducted in our study, including dataset descriptions, model architectures, training procedures, and evaluation metrics.

### D.1. Data Generation Process

We simulate a scenario where there is no confounding in the population but the data is subject to both (1) deterministic (truncation) and (2) non-deterministic selection bias. The data generation process is defined as follows:

**Population Data Generation:** Covariates $X$ are drawn from a uniform distribution, and treatment assignment $T$ follows a linear propensity score model ensuring good population overlap:

$$X \sim \text{Uniform}(-3, 3) \tag{D.1}$$
$$e^*(X) = 0.5 + 0.1X \tag{D.2}$$
$$T \sim \text{Bernoulli}(e^*(X)) \tag{D.3}$$

The potential outcomes $Y(0)$ and $Y(1)$ are generated via non-linear polynomial mean functions with additive noise:

$$Y(t) = \mu_t(X) + \epsilon_t, \quad \text{for } t \in \{0, 1\} \tag{D.4}$$
$$\mu_0(X) = 1.0 + 0.5X - 0.2X^3 \tag{D.5}$$
$$\mu_1(X) = 3.0 - 0.5X + 0.3X^3 \tag{D.6}$$

The noise terms $\epsilon_t$ are drawn from a specified distribution $\mathcal{D}_{\text{noise}}(0, \sigma)$, where we experiment with Normal, Laplace, Log-Normal, and Pareto distributions (centered to zero mean) with scale $\sigma = 0.5$.

**Selection Mechanism:** We observe a sample $(X_i, Y_i, T_i)$ only if the selection indicator $S_i = 1$. The selection mechanism consists of two stages: 1. **Deterministic Truncation:** Control units ($T = 0$) are deterministically unobserved in the tails of the covariate distribution:

$$S_{\text{det}} = \mathbb{I}(T = 1 \vee |X| \leq x_{\text{thresh}}) \tag{D.7}$$

where $x_{\text{thresh}} = 2.0$. 2. **Non-Deterministic Selection:** A probabilistic selection mechanism based on the outcome $Y$ is applied to the remaining samples:

$$P(S = 1 | Y, S_{\text{det}} = 1) = \sigma(\alpha(Y - \gamma)) \tag{D.8}$$

where $\sigma(\cdot)$ is the sigmoid function, $\alpha = 3.0$ controls the selection strength, and $\gamma = 1.5$ is the centering parameter.

| | DAG Framework S-id (Abouei et al., 2024) | Ours |
|---|---|---|
| 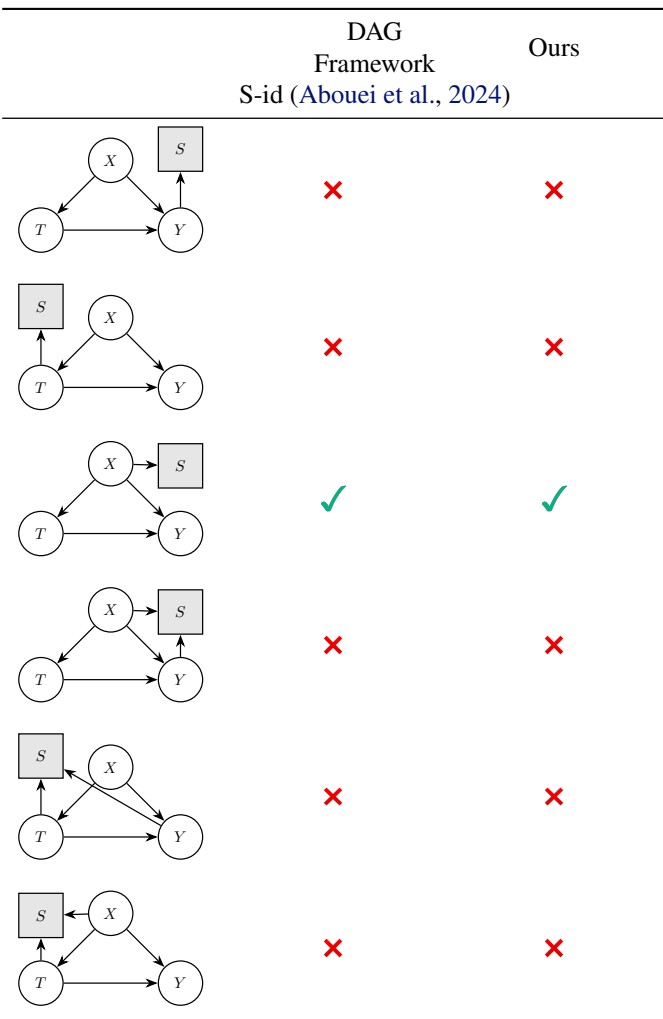 | ✗ | ✗ |
| | ✗ | ✗ |
| | ✓ | ✓ |
| | ✗ | ✗ |
| | ✗ | ✗ |
| | ✗ | ✗ |

*Table 3.* A comparison of S-id identifiablity (Abouei et al., 2024) and identifiablity result under our framework. We are considering ATE identifiability in the sub-population here (s-ATE-sub, Definition 2.3), and the result in different frameworks matched each other in different selection bias cases.

## D.2. Parameterization of Estimators

We specify the estimators that are needed in the following process: (1) estimate propensity $\hat{e}(x)$, (2) identify overlap regions $\mathcal{S}_t$, (3) train models $\hat{\mu}_t$ only on $\mathcal{S}_t$, and (4) extrapolate to the full population $X$.

### D.2.1. PROPENSITY ESTIMATION AND TRUNCATION

The propensity score $\hat{e}(x)$ is estimated using a Multi-Layer Perceptron (MLP) classifier with two hidden layers of 32 units and ReLU activation. The output is calibrated using isotonic regression via 5-fold cross-validation. The valid training sets are defined as:

$$\mathcal{S}_1 = \{i : \hat{e}(X_i) \geq c\}, \quad \mathcal{S}_0 = \{i : \hat{e}(X_i) \leq 1 - c\} \tag{D.9}$$

with truncation threshold $c = 0.05$.

### D.2.2. BASELINE ESTIMATORS

- **IPW:** Estimates the mean via inverse propensity weighting restricted to the overlap region:

$$\hat{\mu}_t^{\text{IPW}}(x) = \frac{\sum_{i \in \mathcal{S}_t} w_i Y_i}{\sum_{i \in \mathcal{S}_t} w_i}, \quad \text{where } w_i = \frac{1}{\hat{P}(T = t | X_i)} \tag{D.10}$$

Note that this estimator computes a constant mean and does not extrapolate conditional expectations.

- **Polynomial Regression:** We model $\mu_t(x)$ using a linear regression on a polynomial basis expansion $\phi(x) = [1, x, x^2, x^3]^\top$:

$$\hat{\mu}_t(x) = \phi(x)^\top \hat{\beta}_t, \quad \hat{\beta}_t = \arg\min_\beta \sum_{i \in \mathcal{S}_t} (Y_i - \phi(X_i)^\top \beta)^2 \tag{D.11}$$

### D.2.3. GAUSSIAN MIXTURE MODEL (GMM) FOR MLE ESTIMATORS

We model the joint density $P(X, Y)$ for each treatment group using a Gaussian Mixture Model with $K = 5$ components:

$$p_t(x, y) = \sum_{k=1}^{K} \pi_k \mathcal{N}\left(\begin{bmatrix} x \\ y \end{bmatrix}; \boldsymbol{\mu}_k, \boldsymbol{\Sigma}_k\right) \tag{D.12}$$

The conditional expectation $\mathbb{E}[Y|X = x]$ is derived analytically from the joint Gaussian parameters.

- **Naive MLE:** Trained via Expectation-Maximization (EM) on the observed data in $\mathcal{S}_t$.

- **MLE + $\beta$** Trained via weighted EM, where sample weights are given by $w_i = 1/\hat{\beta}(X_i, Y_i, T_i)$ (see Selection Model below).

### D.2.4. SCORE MATCHING ESTIMATORS (DEEP ENERGY-BASED MODELS)

We estimate the score function $s_\theta(x, y) \approx \nabla_y \log p(y|x)$ using a neural network $f_\theta(x, y)$.

- **Score Network Architecture:** An MLP with input dimension 2 $(x, y)$, two hidden layers of 64 units, and Softplus activations (to ensure differentiability).

- **Selection Network (for Correction):** To correct for selection bias, we simultaneously learn a selection weight function $\beta_\phi(x, y, t)$ modeled as:

$$\beta_\phi(x, y, t) = \exp\left(\text{MLP}_\phi([x, y, t])\right) \tag{D.13}$$

where $\text{MLP}_\phi$ has one hidden layer of 10 units and Tanh activation.

- **Corrected Objective:** We minimize the Hyvärinen score matching loss on the *composite* score of the observed distribution, $s_{\text{obs}} = s_\theta(x, y) + \nabla_y \log \beta_\phi(x, y, t)$. The loss function is:

$$\mathcal{L}(\theta, \phi) = \mathbb{E}_{\text{obs}}\left[\frac{1}{2}\|s_{\text{obs}}\|^2 + \text{tr}(\nabla_y s_{\text{obs}}) - \lambda_1 \log \beta_\phi + \lambda_2 \|\beta_\phi\|_2^2\right] \tag{D.14}$$

where the term $-\log \beta_\phi$ maximizes the likelihood of selection for observed samples, and $\lambda_1, \lambda_2$ are regularization hyperparameters.

- **Inference:** The conditional expectation is approximated via numerical integration over a grid of $Y$, using the unnormalized density $\hat{p}(y|x) \propto \exp(\int s_\theta(x, y) dy)$.

## D.3. More Details on Training and Evaluation

Once the models are trained on the valid overlap regions, we estimate the Conditional Average Treatment Effect (CATE) for a target unit $x$ by evaluating the difference in conditional expectations: $\hat{\tau}(x) = \hat{\mathbb{E}}[Y(1)|x] - \hat{\mathbb{E}}[Y(0)|x]$. The specific method for computing $\hat{\mathbb{E}}[Y|x]$ depends on the estimator used. We set the learning rate to $0.01$ for all training process and the regularizer $\lambda = 0.05$ for selction corrections. We set the learning rate to $0.01$ for all training process and the regularizer $\lambda = 0.05$ for selction corrections.

D.3.1. SCORE MATCHING ESTIMATOR (NUMERICAL INTEGRATION)

The Score Network $s_\theta(x, y)$ learns the gradient of the log-density, $s_\theta(x, y) \approx \nabla_y \log p(y|x)$. To recover the conditional expectation, we perform numerical integration over a discretized grid of outcome values $\mathcal{Y} = \{y_1, \ldots, y_M\}$ with step size $\Delta y$.

First, we reconstruct the unnormalized log-density $\log \tilde{p}(y_j|x)$ via Riemann summation of the score:

$$\log \tilde{p}(y_j|x) \approx \sum_{k=1}^{j} s_\theta(x, y_k) \Delta y \tag{D.15}$$

We then recover the normalized probability mass function $\hat{p}(y_j|x)$ using the softmax operation over the grid to approximate the partition function $Z(x)$:

$$\hat{p}(y_j|x) = \frac{\exp\left(\log \tilde{p}(y_j|x)\right)}{\sum_{k=1}^{M} \exp\left(\log \tilde{p}(y_k|x)\right)} \tag{D.16}$$

Finally, the conditional mean is computed as the expected value under this discrete distribution:

$$\hat{\mu}(x) = \sum_{j=1}^{M} y_j \cdot \hat{p}(y_j|x) \tag{D.17}$$

D.3.2. GAUSSIAN MIXTURE MODEL (ANALYTICAL SOLUTION)

The GMM estimates the joint density $p(x, y)$ as a mixture of $K$ Gaussian components with parameters $\pi_k, \boldsymbol{\mu}_k, \boldsymbol{\Sigma}_k$. We compute the conditional expectation analytically.

Let the parameters for the $k$-th component be decomposed as $\boldsymbol{\mu}_k = (\mu_{x,k}, \mu_{y,k})$ and variance terms $\sigma_{xx,k}^2, \sigma_{xy,k}$. The conditional expectation of $Y$ given $X = x$ for component $k$ is linear:

$$m_k(x) = \mu_{y,k} + \frac{\sigma_{xy,k}}{\sigma_{xx,k}^2}(x - \mu_{x,k}) \tag{D.18}$$

The marginal likelihood of $x$ under component $k$ is given by the univariate Gaussian:

$$p(x|k) = \mathcal{N}(x; \mu_{x,k}, \sigma_{xx,k}^2) \tag{D.19}$$

The final conditional expectation is the average of the component-wise conditional means, weighted by the posterior probability (responsibility) of each component given $x$:

$$\hat{\mu}(x) = \frac{\sum_{k=1}^{K} \pi_k p(x|k) m_k(x)}{\sum_{k=1}^{K} \pi_k p(x|k)} \tag{D.20}$$

**D.4. Supplementary results**

In this section, we provide additional experimental results to complement the main findings presented in the paper. We include a visualization of the estimated data distributions, as well as extended tables of results for various noise distributions scenarios.

As we can see in Figure 4, the corrected estimators (both MLE and Score Matching) are able to recover the true underlying data distribution more accurately (the estimated curves are closer to the true mean) compared to their naive counterparts, which suffer from bias due to selection effects.

**D.5. More Details on the AoS Dataset**

We generate $Y$ according to the following process: $Y = 0.1X + T \cdot X + \epsilon$. The noise term $\epsilon$ is drawn from a Normal distribution $\mathcal{N}(0, 1)$. The selection mechanism is applied in the same manner as described in Section D.1.

# E. More Experimental Results

We include more results regarding more baselines, functional and parameter scope.

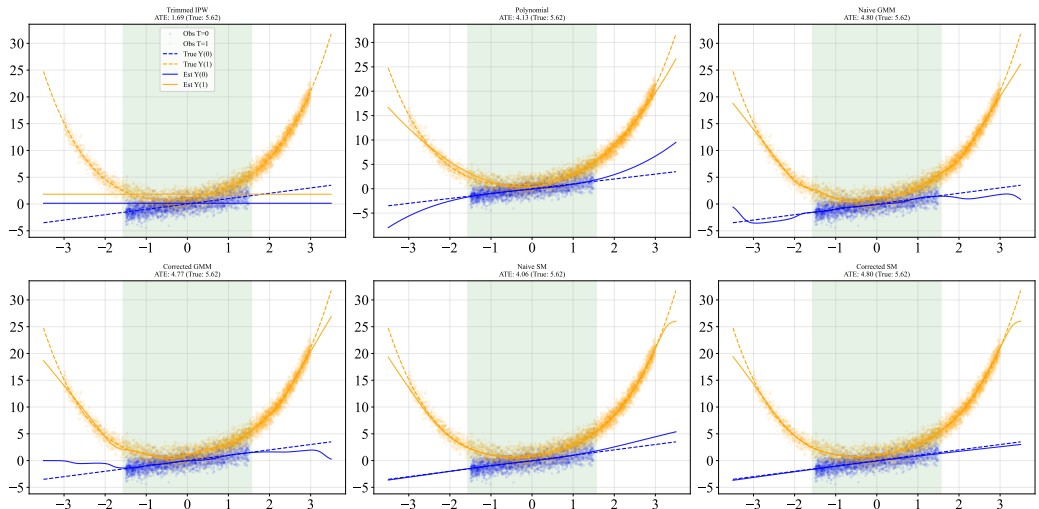

*Figure 4.* Visualization of estimated data distributions.

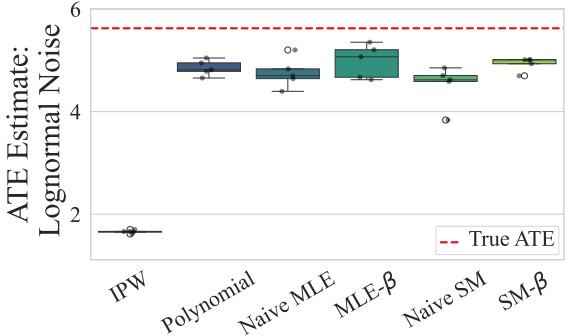

*(a)* ATE Estimates with Log-Normal Noise (Additive)

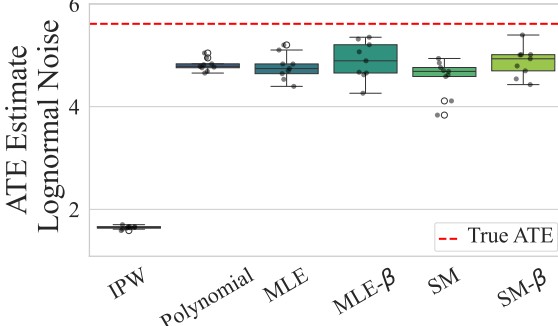

*(b)* ATE Estimates with Log-Normal Noise (Multiplicative)

*Figure 5.* Additional results for Log-Log-Normal noise distribution.

**Added baselines.** AIPW (Bang & Robins, 2005) and TMLE (Van der Laan et al., 2011) (naive sub-population ATE, no selection correction), classical Heckman (Heckman, 1977) correction (jointly Gaussian errors, probit selection model), and AIPW(oracle) on the full unselected dataset as an upper bound.

**Varying selection strength.** We sweep the selection parameters (the sigmoid centering $\beta_C \in \{1.0, 3.0, 5.0\}$ and scaling $\beta_S \in \{0.1, 0.5, 1.0\}$, where $P(S|X, Y, T) = 1/(1+e^{-logits})$, $logits = (Y+0.1X - \beta_C) * \beta_S$) to evaluate the performance of the proposed method. By default, we conduct the experiment with additive Gaussian noise.

**Varying functional form.** We include two additional experiments with non-polynomial, non-monotone functional form *Sin*: $Y = 2X\sin(2X) + T(X^2 + 0.1X^4) + \epsilon$ and *Log*: $Y = X\log(X + 4) + T(X^2\log(2X) + 0.1X^4) + \epsilon$. We apply the same deterministic and nondeterministic selection mechanisms as in the main experiments. This tests whether our estimators MLE+$\beta$ and SM+$\beta$ can handle more challenging functional forms that cannot be modeled well with polynomial estimators.

**Selection depending on X, Y, or both.** Note that we already consider the most challenging case where selection depends on all of T, X, Y simultaneously.

### E.1. Varying Functional Forms

With the Log and Sin functions, all the other methods suffer from selection bias. In particular, while in the main experiments (Figure 2a) where we use polynomial functions, the Polynomial estimator performs relatively well, when we use the Log or Sin function, the Polynomial estimator behaves poorly, while our methods remain robust.

*Table 4.* Results with different functional forms (Mean error, std)

| Method | Sin | Log |
|---|---|---|
| AIPW Oracle | 0.10 (0.06) | 0.04 (0.05) |
| AIPW | 3.61 (0.08) | 3.85 (0.11) |
| TMLE | 3.58 (0.07) | 3.70 (0.12) |
| Heckman | 4.25 (0.09) | 4.26 (0.08) |
| IPW | 3.62 (0.08) | 4.34 (0.07) |
| Polynomial | 3.66 (0.15) | -2.58 (0.39) |
| MLE | 2.04 (0.11) | -3.71 (0.32) |
| MLE+$\beta$ | 3.63 (1.76) | 1.66 (3.24) |
| SM | 2.87 (1.11) | -1.65 (0.84) |
| SM+$\beta$ | **1.28** (1.49) | **0.31** (3.47) |

### E.2. Varying Selection Strength

Under varying selection strength, our methods which take selection bias into account remain robust.

*Table 5.* Results with varying selection strength (Mean Error, std)

| Method | C1.0/S0.1 | C1.0/S0.5 | C1.0/S1.0 | C3.0/S0.1 | C3.0/S0.5 | C3.0/S1.0 |
|---|---|---|---|---|---|---|
| AIPW Oracle | 0.05 (0.02) | 0.05 (0.02) | 0.05 (0.02) | 0.05 (0.02) | 0.05 (0.02) | 0.05 (0.02) |
| AIPW | 3.76 (0.03) | 3.84 (0.03) | 4.03 (0.03) | 3.72 (0.03) | 3.81 (0.05) | 3.99 (0.07) |
| TMLE | 3.76 (0.03) | 3.81 (0.03) | 3.99 (0.03) | 3.72 (0.03) | 3.75 (0.05) | 3.94 (0.08) |
| Heckman | 3.79 (0.03) | 4.02 (0.04) | 4.34 (0.04) | 3.75 (0.04) | 3.90 (0.04) | 4.26 (0.07) |
| IPW | 4.10 (0.04) | 4.16 (0.03) | 4.17 (0.03) | 4.08 (0.03) | 4.13 (0.03) | 4.21 (0.07) |
| Polynomial | 0.45 (0.14) | 0.13 (0.13) | **0.17** (0.16) | 0.34 (0.03) | $-0.86$ (0.08) | $-1.35$ (0.18) |
| MLE | **0.08** (0.14) | $-0.18$ (0.14) | 0.30 (0.21) | **0.15** (0.14) | $-0.81$ (0.20) | $-1.98$ (0.42) |
| SM | 0.60 (0.18) | 0.29 (0.19) | 0.20 (0.21) | 0.33 (0.30) | $-0.76$ (0.25) | $-2.90$ (3.45) |
| MLE+ | $-0.10$ (0.28) | $-$**0.03** (0.55) | 0.83 (0.83) | 0.14 (0.13) | $-0.60$ (0.82) | $-1.76$ (0.76) |
| SM+ | 0.40 (0.21) | 0.20 (0.13) | 0.20 (0.17) | 0.24 (0.69) | $-$**0.23** (0.65) | $-$**1.10** (0.33) |

