# OpenReview forum: "Towards a Holistic Understanding of Selection Bias for Causal Effect Identification"
_ICML.cc/2026/Conference — ICML 2026 regular_

### Official Review · Reviewer_4sDt · 2026-03-05

**Soundness:** 4
**Presentation:** 3
**Significance:** 4
**Originality:** 4
**Overall Recommendation:** 5
**Confidence:** 5

**Summary:**

This paper proposes identifiability conditions for the average treatment effect (ATE) in the presence of selection bias. Moving beyond the limitations of existing approaches that rely on DAGs, the authors prove necessary and sufficient conditions showing that ATE can be identified in more general settings (that is, across a wider range of selection bias scenarios) by leveraging assumptions on distribution classes.

**Compliance With Llm Reviewing Policy:**

Affirmed.

**Final Justification:**

I thank the authors for their comprehensive and well-organized rebuttal. My concerns have been fully addressed, and I remain confident in my accept recommendation.
My initial concern regarding the generality of the distributional assumptions has been satisfactorily clarified.
The expanded experimental evaluation also substantially addresses my empirical concerns. In all evaluated conditions, the proposed approach achieves the lowest error, further reinforcing the paper's empirical claims.
Overall, I am confident that this is a strong theoretical contribution with rigorous foundations and sufficient empirical support, and I maintain my accept score.

**Key Questions For Authors:**

- Beyond standard IPW and regression adjustment, numerous sophisticated ATE estimation methods have been developed in recent years. At a minimum, comparisons against methods such as TMLE would seem warranted rather than relying on basic MLE. Is there a specific reason for the choice of baseline competitors?
- How does performance change when using multivariate covariates X (e.g., at least 5 dimensions) rather than the univariate setting? Additionally, what are the results when selection depends not only on Y but also on X, or jointly on both X and Y?

**Limitations:**

- While the novel identifiability conditions are theoretically rigorous and intellectually compelling, the restriction to specific distribution classes is a notable limitation.
- The balance between the theoretical contributions and the empirical evaluation could be improved. A more diverse set of experiments(e.g., varying in dimensionality, selection mechanisms, and data complexity) would strengthen the paper considerably.

**Strengths And Weaknesses:**

Strengths
- Proposing conditions that are more general than existing backdoor criteria holds significant value in the causal inference literature.
- The theoretical foundations are notably rigorous and well-developed.

Weaknesses
- The proposed estimation methods are ultimately restricted to distribution classes satisfying Proposition 3.3 or 3.5. While the identifiability results are made more general, the range of distributions for which they hold is constrained (effectively creating a trade-off). Given this trade-off, to what extent can these conditions truly be considered "general"?

---

> ### Author Rebuttal · Authors · 2026-03-31
>
> **Q1: Generality of distributional assumptions.**
>
> The reviewer raises a fair point about the generality of Condition 1 and the limitation to the specific distribution classes in Propositions 3.3 and 3.5. We want to make several clarifications.
>
> 1. Propositions 3.3 and 3.5 represent **sufficient** **conditions (which are not exhaustive)** that we have verified for common distributional families, but Condition 1 itself is the fundamental general characterization. Any distribution class for which the density ratio argument holds (i.e., two distributions with different means produce unbounded density ratios in the tails) will satisfy Condition 1. The four families in Proposition 3.5  were chosen because they span a broad spectrum of tail behaviors (light-tailed and heavy-tailed) and they are widely used in applied work. Similarly, the log-polynomial class in Proposition 3.3 is quite rich, encompassing Gaussian, exponential-family, and polynomial-tailed distributions.
> 2. **Distributional assumptions are standard.** We kindly refer the reviewer to **R2(5JGq) Q1** for a detailed comparison.
>
> **Q2: Choice of baselines and more extensive experiments**
>
> We thank the reviewer for the suggestion. We **include more baselines** (please see **R3(4rWp) Q4** for details) and evaluate on a **broader scope of parameters**, and **new applicable settings with high-dimensional covariates** to further address the reviewer's concern on experiments:
>
> **Varying selection strength:** we sweep the selection parameters (the sigmoid centering $\beta_C\in 1.0, 3.0$ and scaling $\beta_S\in 0.1, 0.5, 1.0$).
>
> **Varying functional form (Sin and Log):** we include non-polynomial, non-monotone forms. Sin: $Y = 2X\sin(2X)+ T(X^2+0.1X^4)+\epsilon$ and Log: $Y=X\log (X+4)+ T(X^2 \log (2X)+0.1X^4)+\epsilon$.
>
> **Selection depending on X, Y, or both:** In the paper and all additional experiments, selection is a function of **all variables** and **both** deterministic and non-deterministic selection are applied simultaneously which is the most challenging scenario.
>
> **Table 1 (Varying selection)**: Mean Error (std)
>
> ||C1.0/S0.1|C1.0/S0.5|C1.0/S1.0|C3.0/S0.1|C3.0/S0.5|C3.0/S1.0|
> |-|-|-|-|-|-|-|
> |IPW|4.10 (0.04)|4.16 (0.03)|4.17 (0.03)|4.08 (0.03)|4.13 (0.03)|4.21 (0.07)|
> |Polynomial|0.45 (0.14)|0.13 (0.13)|0.17 (0.16)|0.34 (0.03)|-0.86 (0.08)|-1.35 (0.18)|
> |AIPW Oracle|0.05 (0.02)|0.05 (0.02)|0.05 (0.02)|0.05 (0.02)|0.05 (0.02)|0.05 (0.02)|
> |AIPW|3.76 (0.03)|3.84 (0.03)|4.03 (0.03)|3.72 (0.03)|3.81 (0.05)|3.99 (0.07)|
> |TMLE|3.76 (0.03)|3.81 (0.03)|3.99 (0.03)|3.72 (0.03)|3.75 (0.05)|3.94 (0.08)|
> |Heckman|3.79 (0.03)|4.02 (0.04)|4.34 (0.04)|3.75 (0.04)|3.90 (0.04)|4.26 (0.07)|
> |MLE|0.08 (0.14)|-0.18 (0.14)|0.30 (0.21)|0.15 (0.14)|-0.81 (0.20)|-1.98 (0.42)|
> |SM|0.60 (0.18)|0.29 (0.19)|**0.20** (0.21)|0.33 (0.30)|-0.76 (0.25)|-2.90 (3.45)|
> |MLE+$\beta$|-0.10 (0.28)|**-0.03** (0.55)|0.83 (0.83)|**0.14** (0.13)|-0.60 (0.82)|-1.76 (0.76)|
> |SM+$\beta$|0.40 (0.21)|0.20 (0.13)|**0.20** (0.17)|0.24 (0.69)|**-0.23** (0.65)|**-1.10** (0.33)|
>
> **Table 2 (Varying function)**: Mean Error (std)
>
> ||Sin|Log|
> |-|-|-|
> |AIPW Oracle|0.10 (0.06)|0.04 (0.05)|
> |AIPW|3.61 (0.08)|3.85 (0.11)|
> |TMLE|3.58 (0.07)|3.70 (0.12)|
> |Heckman|4.25 (0.09)|4.26 (0.08)|
> |IPW|3.62 (0.08)|4.34 (0.07)|
> |Polynomial|3.66 (0.15)|-2.58 (0.39)|
> |MLE|2.04 (0.11)|-3.71 (0.32)|
> |MLE-$\beta$|3.63 (1.76)|1.66 (3.24)|
> |SM|2.87 (1.11)|-1.65 (0.84)|
> |SM-$\beta$|**1.28** (1.49)|**0.31** (3.47)|
>
> **New applicable setting with high-dimensional covariates:** We provide new Gnome-Wide-Association Studies (GWAS) and Mendelian Randomization Studies (MR) experiments, expanding to high-dimensional, biologically motivated settings. **The theoretical guarantee** under these two settings can be extended from our main Theorem 3.1, and we will add them in the revision.
>
> - **GWAS (Table 3)**, where we estimate the genetic variant G which associate with multiple phenotypes $Y_i$. 100 genetic variants (genotypes) and 3 phenotypes.
> - **MR (Table 4),** where we use genotype G as instrumental variables to estimate treatment effects under hidden confounding.100 genetic variants (genotype),  3 observed covariates, and 1 unobserved confounder $U$, where selection induces a spurious $G–U$ association.
>
> **Table 3 (GWAS results)**: MSE (x$10^{-2}$, std) of the estimated effect size matrices.
> |Noise|Normal|Laplace|
> |-|-|-|
> |Covariance|10.07 (0.01)|10.09 (0.01)|
> |MLE|0.59 (0.00)|1.39 (0.01)|
> |SM|0.56 (0.00)|1.33 (0.00)|
> |MLE+$\beta$|0.54 (0.00)|1.36 (0.00)|
> |SM+$\beta$|**0.34** (0.00)|**0.89** (0.00)|
>
> **Table 4 (MR results):** Mean Error (std)
> |Noise|Normal|Laplace|
> |-|-|-|
> |Ideal 2SLS (Full Pop)|-0.02 (0.04)|-0.04 (0.07)|
> |Naive 2SLS (Biased)|3.97 (0.63)|7.83 (2.10)|
> |MLE|-0.11 (0.31)|-1.22 (0.71)|
> |SM|0.21 (0.41)|0.04 (0.94)|
> |MLE+$\beta$|**-0.03** (0.37)|-1.26 (0.64)|
> |SM+$\beta$|0.56 (0.45)|**-0.02** (0.66)|
>
>  In all settings, our method achieve the lowest error. We believe the diverse evaluation strengthens our paper.

---

> > ### Author Rebuttal · Reviewer_4sDt · 2026-03-31
> >
> > The authors effectively clarified that the distributional assumption I had previously held was somewhat strong. After reviewing this rebuttal, I feel more assured in the accept score I had assigned, and my overall confidence in that decision has grown.

---

### Official Review · Reviewer_4rWp · 2026-03-10

**Soundness:** 3
**Presentation:** 3
**Significance:** 3
**Originality:** 3
**Overall Recommendation:** 5
**Confidence:** 4

**Summary:**

In this paper, the authors propose a general identification framework for truncated data under selection bias. Based on the identification results, they also propose a general ATE estimation framework for data that satisfy the identification conditions. Experiments on synthetic and semi-synthetic data demonstrate the effectiveness of the proposed framework in many truncation scenarios.

**Compliance With Llm Reviewing Policy:**

Affirmed.

**Final Justification:**

All of my concerns are resolved, and I have raised my score to 5.

**Key Questions For Authors:**

Please see weaknesses.

**Limitations:**

Yes.

**Strengths And Weaknesses:**

**Strengths**

1. While many works focus on causal effect estimation under selection bias in censored data [1-8], only a few address ATE estimation under selection bias in truncated data. This paper provides new insights into the latter field.

2. The causal graphs of selection bias cover most possible scenarios, thereby making the results in this paper applicable to many truncated datasets.

3. The theoretical results are sound.

**Weaknesses**

1. The positioning of this paper could be reconsidered. As stated in Heckman's selection bias framework [9, 10], there are two scenarios where selection bias occurs: censoring and truncation. Many previous works have studied selection bias in censored data, providing comprehensive identification and estimation results across all the scenarios discussed in Table 1 under more relaxed assumptions than those made in this paper or the related works discussed [1-8]. In contrast, this paper focuses on selection bias in truncated data, where no external data are available for the unselected units, thereby requiring additional parametric or distributional assumptions to make the ATE identifiable. I understand that these works for censored data may be out of scope for this paper, as it focuses on truncated data. However, the authors could at least make it clear in the title, abstract, introduction, and problem formulation that it is not "a holistic understanding of" all kinds of selection bias, but instead is "a holistic understanding of" selection bias in truncated data.

2. The assumptions needed by the proposed identification are not "weak". In fact, such parametric or distributional assumptions could be easily violated in complex real-world scenarios, and are mostly untestable. Additionally, there are also similar parametric or distributional assumptions made by previous works that focus on each scenario in Table 1, but not all of them [11, 12]. I think a better choice is to state that these assumptions are "weaker" than existing "general" identification frameworks.

3. Experiments are conducted on only synthetic and semi-synthetic data, making it difficult to assess the effectiveness of the proposed method in complex real-world applications. The authors could conduct experiments on real-world datasets to enhance the persuasiveness of their results.

4. Only two baselines are compared, and more baselines for truncated data should be considered.

**References**

[1] Infante-Rivard, Claire, and Alexandre Cusson. "Reflection on modern methods: selection bias—a review of recent developments." International journal of epidemiology 47.5 (2018): 1714-1722.

[2] Colnet, Bénédicte, et al. "Causal inference methods for combining randomized trials and observational studies: a review." Statistical science: a review journal of the Institute of Mathematical Statistics 39.1 (2024): 165.

[3] Horvitz, Daniel G., and Donovan J. Thompson. "A generalization of sampling without replacement from a finite universe." Journal of the American statistical Association 47.260 (1952): 663-685.

[4] Rotnitzky, Andrea, James M. Robins, and Daniel O. Scharfstein. "Semiparametric regression for repeated outcomes with nonignorable nonresponse." Journal of the american statistical association 93.444 (1998): 1321-1339.

[5] Little, Roderick JA, and Donald B. Rubin. Statistical analysis with missing data. John Wiley & Sons, 2019.

[6] Tchetgen Tchetgen, Eric J., and Kathleen E. Wirth. "A general instrumental variable framework for regression analysis with outcome missing not at random." Biometrics 73.4 (2017): 1123-1131.

[7] Li, Wei, Wang Miao, and Eric Tchetgen Tchetgen. "Non-parametric inference about mean functionals of non-ignorable non-response data without identifying the joint distribution." Journal of the Royal Statistical Society Series B: Statistical Methodology 85.3 (2023): 913-935.

[8] Li, Baohong, et al. "Learning shadow variable representation for treatment effect estimation under collider bias." Forty-first International Conference on Machine Learning. 2024.

[9] Heckman, James J. "The common structure of statistical models of truncation, sample selection and limited dependent variables and a simple estimator for such models." Annals of economic and social measurement, volume 5, number 4. NBER, 1976. 475-492.

[10] Heckman, James J. "Sample selection bias as a specification error." Econometrica: Journal of the econometric society (1979): 153-161.

[11] Wooldridge, Jeffrey M. Econometric analysis of cross section and panel data. MIT press, 2010.

[12] Wang, Yuyao, Andrew Ying, and Ronghui Xu. "Doubly robust estimation under covariate-induced dependent left truncation." Biometrika 111.3 (2024): 789-808.

---

> ### Author Rebuttal · Authors · 2026-03-31
>
> **Q1: The distinction between censored and truncated data**
>
> We thank the reviewer for the careful and insightful feedback. After careful reflection, we address this question with two points and hope to make the positioning clearer:
>
> 1. **Positioning: censored vs. truncated data.**
>
>  We agree that our setting falls squarely within the *truncation* regime in Heckman's terminology: non-selected units are entirely absent from the dataset, and we observe $(X_i,Y_i,T_i)$ only for individuals with $S_i=1$. **This applies to both our deterministic selection** (where region $\mathcal{B}$ is completely unobserved) **and nondeterministic selection** (where data points are probabilistically excluded). In contrast, Heckman's *censored* setting assumes that certain covariates are observed for all individuals, including non-selected ones. **Our paper does not assume access to any data from non-selected units. This is the realistic setting in many applied contexts and arguably harder**, e.g. in UK Biobank, non-participants are entirely absent from the dataset; there is no covariate information for individuals who chose not to volunteer. The same holds for survey non-response, electronic health records (where non-care-seekers are invisible). We will include the literature suggested [1-8] by the reviewer on censored data setting to make this positioning explicit in the revision.
>
> 2. **Relation to missing data problem in causal inference literature** (Rather than selection bias)
>
> Inspired by this question, we also reflect on the connection of this problem to the broader literature and note that Heckman's "censoring" is commonly referred to as **missing data** **in the modern causal inference and statistics literature**. This connection was formalized by Rubin's missing data framework (Rubin, 1976; Little & Rubin, 2002), which classifies missingness mechanisms into three categories: missing completely at random (MCAR), missing at random (MAR), and missing not at random (MNAR). **Under the MAR assumption, where the probability of missingness depends only on observed variables** is the logic underlying Heckman's *censoring* paradigm.
>
> Note that, in the paper, we also distinguish a deterministic (also stated as hard truncation, Proposition 3.3) and non-deterministic selection mechanism (soft selection, Proposition 3.5). The truncation here means not only truncation in the Heckman’s sense, but also that the entire region of the$ (X, Y, T)$ space are unobserved, i.e. $P(S=1|V) = 0/ 1$ , while the soft selection refers to the cases when every point in the support still have positive probability of being observed, i.e. $0 < P(S=1|V) < 1$.
>
>
>
> **Q2: Assumptions are not "weak" but should be "weaker than existing general frameworks."**
>
> We agree with this more precise characterization and will revise the language throughout the paper accordingly. Specifically, we will replace claims of "weak assumptions" with **"weaker assumptions than those required by existing general identification frameworks."**  We also kindly point the reviewer to our answer to **R2(5JGq) Q1** for a thorough analysis on the comparison of graphical and distributional assumptions that we made.
>
>
>
> **Q3: Experiments in real-world applications.**
>
> We appreciate this suggestion. We have conducted new experiments in two biologically motivated settings that go substantially beyond the original synthetic and semi-synthetic evaluations and kindly refer the reviewer to **R4(4sDt) Q2** for the results.
>
>
>
> **Q4: More baselines for truncated data should be considered.**
>
> We agree that the original comparison with only IPW and polynomial regression is insufficient. In the revised experiments, we have expanded the baseline set.
>
> **Adding baselines:** We add the doubly robust estimator (AIPW) (Robins, 1994) and TMLE (van der Laan & Rose, 2011) to estimate the sub-population ATE without correcting for selection. We also include the classical Heckman (1979) correction, which assumes jointly Gaussian errors and a parametric probit selection model. Lastly, we use AIPW(oracle) for an oracle estimation on the unselected dataset for an upper bound of all the estimators. We expand our experiments to include those baselines and with varying selection strength and functional forms. **The key takeaway is that** the naive IPW (mean = 1.65) in the censored dataset is dramatically biased VS the ground truth ATE (= 5.6), while the estimation methods that we proposed **(MLE+$\beta$, SM+$\beta$) substantially reduce the bias** (mean = 5.34, 5.17), outperforming their uncorrected counterparts (MLE:5.01, SM:4.78). Polynomial estimator specifically targets the cases where Y is a polynomial function of X, which is satisfied in our simulation setup, while other estimation methods (MLE, MLE+$\beta$, SM, SM+$\beta$) can generalize beyond polynomial function classes. We kindly refer the reviewer to **R4(4sDt) Q2** for the complete result and apoligize for the inconvenience due to space limit.

---

> > ### Author Rebuttal · Reviewer_4rWp · 2026-04-01
> >
> > Thank the authors for their detailed response, as well as their efforts in reviewing a substantial body of literature on missing data and censoring, and conducting additional experiments. All of the concerns have been fully addressed. I will raise my score to 5, indicating a clear accept.

---

### Official Review · Reviewer_5JGq · 2026-03-11

**Soundness:** 2
**Presentation:** 2
**Significance:** 2
**Originality:** 3
**Overall Recommendation:** 4
**Confidence:** 4

**Summary:**

The paper proposes a generalized framework for ATE estimation under selection bias. Specifically, the paper focuses on a setting in which the sample observed is selected according to some unobserved mechanism, and therefore its distribution may be different from that of the target population of interest. The paper established a condition for identifiability of the ATE in this setting that is based on imposing what are essentially shape restrictions on the allowed probability families for the data. These shape restrictions make it possible to estimate selection probabilities, and thus correct for them in downstream analyses even when the selection indicator is not observed, therefore rendering the ATE identifiable. The condition is shown to be general in that it encompasses many of the existing conditions for ATE identification in the literature on graphical criteria.

**Compliance With Llm Reviewing Policy:**

Affirmed.

**Final Justification:**

My final recommendation is a weak accept. I believe that the author's rebuttal presented convincing arguments to lessen the concerns that I had originally presented in my review.

**Key Questions For Authors:**

1. Can you provide examples of actual applied setting in which the assumptions you propose would be better than assuming something about the location of the selection?
2. Can you provide some ways to compute uncertainty around your estimates? For the method to be usable analysts need to be able to compute uncertainty.

**Limitations:**

Yes.

**Strengths And Weaknesses:**

Soundness: The submission is technically sound. The theoretical analysis is particularly thorough, while the experimental evaluation could use some small addition but is also convincing. Specifically it would be good to see some baselines (e.g., an oracle estimator with the selection observed). In addition no theoretical statement or even informal discussion of the large-sample behavior of the proposed estimators or of uncertainty quantification is provided, and this reduces the soundness of the proposed approach. The paper also presents its proposed conditions as weaker than some (or all?) of the existing conditions for ATE identification. I don't think what we see in the paper substantiates this claim.

Presentation: The submission is well written. however the presentation as it is currently makes it quite hard to understand exactly how the proposed methodology achieves what it claims to. Specifically it isn't immediately clear from the presentation what assumptions are leveraged in order to achieve identification in settings that (according to the paper) are not currently covered by the DAG or SEM frameworks. The paper should be clearer in explaining upfront what the restrictions that allow for inference are, and how they are used.

Significance: The identification condition proposed by the paper is interesting and (to my knowledge) novel. However I have severe doubts about its wide applicability in practical causal inference scenarios. Specifically, as the paper notes, current methods focus on understanding where (in the space of the variables) the selection is happening, while the proposed method bypasses the issue by focusing on restricting the observed distributions. There are some very good reasons for why existing ATE estimation approaches focus on pinpointing where selection occurs, rather than making assumptions about the distributions under study: they do this because, in practice, analysts do not want to make assumptions about the population distribution that they study (e..g, that they are normal or some other known family) and instead prefer making assumptions about the processes they study: this is, largely, because it is easier to collect data or existing literature to corroborate these assumptions than it is to corroborate assumptions about the shape of some unknown distribution. In practice the proposed method seems to be applicable to a small subset of parametric families (even though the condition for identification is quite general) and I am unsure that analysts will ever want to make these parametric assumptions in practice.

Originality: The work is original to the best of my knowledge in that it provides a novel identification condition for the ATE and studies it thoroughly.

---

> ### Author Rebuttal · Authors · 2026-03-31
>
> **Q1: Practical applicability of distributional assumptions vs. graphical assumptions.**
>
> We thank the reviewer for this thoughtful feedback. The reviewer correctly observes that sometimes structural assumptions are prefered over distributional ones. We make several points in response:
>
> **1. Real-world settings where our assumptions are natural.** In large-scale Genome Wide Association Studies (GWAS), **distributional (Gaussian) assumptions are pervasive** (Loh et al., 2015; Mbatchou et al., 2021; Bulik-Sullivan et al., 2015; Lloyd-Jones et al., 2019), precisely the families covered by Prop 3.3 and 3.5. **In contrast,** it is often unknown which variables the selection depends on, as participants self-select based on complex combinations of traits, thus hard to specify the correct DAG, while **distributional assumptions can be empirically assessed** on the observed data. In Mendelian Randomization (MR) studies, existing work (Schoeler et al., 2023) assumes the selection is based on X or Y, but not on the instrument G or jointly on all variables. We **cover all cases** without assumption on selection location.
>
> **2. Distribution assumptions are also common in causality** (Shimizu et al., 2006; Peters et al., 2014). The classical Heckman selection model (Heckman, 1979) assumes jointly Gaussian error, and (Zhang et al, 2016) make assumptions on the noise function class as well as distributions (Non-Gaussian noise), and limited to outcome-dependent selection.
>
> **3. New GWAS and MR experiments.** We kindly point the reviewer to **R4(4sDt) Q2** for the results and apologize for this inconvenience.
>
> **Q2: Uncertainty quantification of the estimator**.
>
> We agree that the gap between our identifiability theory (Theorem 3.1) and the estimation procedure (Algorithm 1) should be formally bridged, therefore we provide consistency guarantees in two-fold:
>
> 1. **Robust Identifiability (Informal Condition 1-R).** The distribution classes triple satisfy Condition 1-R with mass function $M: (0,\infty) \to [0,1]$ if, for any compatible tuples $(P\_{t|x}, P\_{xy(t)}, P\_{s|xyt})$ and $(Q\_{t|x}, Q\_{xy(t)}, Q\_{s|xyt})$ with $|\tau\_{P\_{xy(t)}} - \tau\_{Q\_{xy(t)}}| > \varepsilon$, there exists a set $\mathcal{A}\_\varepsilon \subseteq \mathcal{A}$ with $P\_{xy(t)}(\mathcal{A}\_\varepsilon), Q\_{xy(t)}(\mathcal{A}\_\varepsilon) \geq M(\varepsilon)/c$ such that for all $(x,y,t) \in \mathcal{A}\_\varepsilon$:
>
> $$\frac{\alpha\_P(x,y,t) \cdot P\_{t|x} \cdot P\_{xy(t)}}{\alpha\_Q(x,y,t) \cdot Q\_{t|x} \cdot Q\_{xy(t)}} \notin \left(\frac{1}{2}, 2\right).$$
>
> This ensures ATE differences are **detectable with non-negligible mass** in the observed data. Condition 1-R reduces to Condition 1 in the limit $\varepsilon \to 0$ and $M(\varepsilon) \to 0$.
>
> 2. **Convergence and asymptotic consistency for MLE and SM estimator**
>
> Given the above Condition 1-R, we give a result that shows such a converging algorithm exists, with our Algorithm 1 an instantiation.
>
> **Informal Theorem.** Fix $c \in (0, 1/2), \varepsilon, \delta \in (0,1)$, and a reference measure $\mu$ over $\mathbb{R}^d \times \mathbb{R}$. Let the concept classes  ($\mathbb{P}\_{t|x}(c), \mathbb{P}\_{xy(t)}, \mathbb{S}$) satisfy *Condition 1-R* with mass function $M(\cdot)$ and some other properties we omit for brevity, then there exists an algorithm that, given $N$ i.i.d. samples from $P(V \mid S=1)$, outputs $\hat{\tau}$ such that with probability $\geq 1 - \delta$: $|\hat{\tau} - \tau_P| \leq \varepsilon$.
>
> **Corollary (Consistency).** Under the conditions of the above theorem, as $n \to \infty, \hat{\tau} \xrightarrow{p} \tau_P$.
>
> Our MLE $+\beta$ and SM $+\beta$ estimators are computationally efficient approximations of this procedure. For MLE$+\beta$, when the parametric family is correctly specified and has finite VC complexity, the MLE is known to achieve the optimal rate for density estimation in L1-norm (Massart & Nédélec, 2006). For SM, under Prop 3.3 and 3.5, it is consistent for the true score function (Hyvärinen & Dayan, 2005). Combined with the extrapolation property of truncated statistics (Daskalakis et al, 2021), this ensures that the estimated density converges to the true density in L1.
>
>
> **Q3: What assumptions enable identification beyond DAG/SEM frameworks?**
>
> We appreciate this feedback. The key insight is that **we formulate assumptions directly on the distributions rather than the DAGs that entail them (which requires additional assumptions for Markovianity and Faithfulness)**. For nondeterministic selection, the tail-divergence property of the families in Prop 3.5 ensures that different ATEs produce observably different selected distributions (Condition 1, ③), even with unknown selection mechanism. For deterministic selection, log-polynomial densities (Prop 3.3) are uniquely determined by their restriction to any positive-measure subset via the extrapolation principle from truncated statistics.
> We will add a discussion section to make it explicit.

---

> > ### Author Rebuttal · Reviewer_5JGq · 2026-04-03
> >
> > I thank the authors for engaging with my review. I appreciate the theoretical work done here and as such I will rase my score.

---

### Official Review · Reviewer_kGhY · 2026-03-19

**Soundness:** 3
**Presentation:** 2
**Significance:** 2
**Originality:** 2
**Overall Recommendation:** 2
**Confidence:** 3

**Summary:**

This paper considers the problem of estimating the average treatment effect  (ATE) with respect to a population given data from a selected subpopulation. Under the assumption of unconfoundedness and overlap, the paper builds on previous work to propose a condition on the class of propensity scores, covariate-outcome distributions and selection mechanism for ATE identifiability. The paper proposes a few general cases under which this condition holds when the selection mechanism is deterministic and otherwise. It also compares with existing work in the literature to show that the proposed condition holds in those cases. The method is evaluated on synthetic data and semi-synthetic data.

**Compliance With Llm Reviewing Policy:**

Affirmed.

**Key Questions For Authors:**

1. Definition of an observation study is inconsistent -  defined in Line 104 with respect to a dataset $\mathcal{D}$, Definition 2.3 and the formal definition of realizability uses an observational distribution $\mathcal{P}\left(\mathbf{V}\right)$. Are these interchangeable?
2. It is unclear to me what the experiments say about the proposed estimation procedure - For example, the performance of the MLE+\beta comparable to polynomial regression in the multiplicative noise case and the variance is larger for the MLE + \beta case. In the case of the semi-synthetic data the difference among any of the methods is negligible with the proposed methods having higher variance than the naive methods.

**Limitations:**

yes

**Strengths And Weaknesses:**

Identification conditions and estimating the ATE is of significant interest to a lot of different practical applications. Under selection bias, works that consider estimating the ATE often have specific assumptions about the selection mechanism. This paper considers a novel approach in that sense since its main contribution, i.e., a condition that is necessary and sufficient for ATE identifiability, is in terms of distribution classes.

1. My main concern is that I think Condition 1 is just the contrapositive of the identifiability definition. The identifiability definition states that if two models agree on the observed distribution they induce, they must agree on the parameter of interest (ATE). In my understanding Condition 1 states that if the ATE is different for two distributions, then they differ on the observed distribution (under selection) which is just a contrapositive of the previous statement. In that sense Theorem 3.1 is direct. Perhaps then the main contribution is Proposition 3.3 and 3.5? Also, Proposition 3.5 has a statement that depends on $d$ yet the condition does not?
2. This work draws heavily from [1] since the form of the condition is similar to [1]. I would expect some text contrasting this current work from that both in terms of setting (which is ofcourse obvious), but also in terms of techniques. The paper often mentions that the reason for the form of the condition 1 is that techniques from truncated statistics can be used, but there is no concrete connection drawn in the main text.
3. The presentation of this paper needs significant improvement. There are typos and notational inconsistencies that making it confusing to read the paper. It's even more confusing because the textual description of compatibility in Section 2 does not match with the formal definition in Section B.1. Often notation is introduced without defining it - for example, $\hat{\beta}$ for the learned selection bias function is not defined.

[1] - Cai, Y., Kalavasis, A., Mamali, K., Mehrotra, A., & Zampetakis, M. (2025, July). What Makes Treatment Effects Identifiable? Characterizations and Estimators Beyond Unconfoundedness. In The Thirty Eighth Annual Conference on Learning Theory (pp. 755-756). PMLR.

---

> ### Author Rebuttal · Authors · 2026-03-31
>
> **Q1: Condition 1 is just the contrapositive of the identifiability definition. The contribution is Prop 3.3 and 3.5.**
>
> We appreciate this observation. We clarify some important subtleties and will incorporate the discussion into the main text:
>
> - **Non-triviality of Condition 1 and Theorem 1**: The reviewer is correct that Condition 1 has the logic of a contrapositive of identifiability. However, **Condition 1 specify the property of the distribution classes** $(\mathbb{P}\_{t|x}, \mathbb{P}\_{xy(t)}, \mathbb{S})$,  unlike Definition 2.2 which characterize all distributions realizable w.r.t. the distribution class triple. **The non-triviality of Theorem 3.1 lies in establishing the necessity and sufficiency** of Condition 1 for s-ATE-full identifiability. Sufficiency requires constructing a well-defined mapping from selected distributions ($\mathcal{P}^{S=1}$) to ATE values, and necessity requires constructing explicit counterexamples when the condition fails. The proof technique for both sufficiency and necessity is not immediate from the definition. See Q2 for details.
> - The reviewer is correct that **our main contributions are Prop 3.3 and 3.5**. All existing identifiability results are subject to Condition 1, while Prop 3.3 and 3.5 are **testable statements** for the distributions where s-ATE-full are identifiable. These propositions **provide new identifiability guarantees on settings that previously considered non-identifiable** (e.g. outcome-dependent selection with Gaussian noise).
>
> **Q2: Contrast to [1] in setting and technique.**
>
> The setting we are concerned with and [1] are **two different problems**. While [1] focuses on overlap violations without selection bias, we address selection bias that distorts the observed distribution. Our condition 1 inherits the language of the identifiability condition for ATE without selection that was developed in [1] to **cast all existing methods into a single framework** and provide **new identifiability guarantees**.
>
> **Three differences**:
>
> - **Novel setting:** Our setting is in general more challenging because we include the unknown selection probability $P(S=1\mid T, X, Y)$, and we need to control this extra degree of freedom absent in [1]’s condition, which only characterizes the joint distribution $P(T, X, Y)$.
>
> - **Proof differences:
>   (a) When selection is not deterministic**, the observed distribution is an (unknown) soft reweighted version of the true distribution, which is unrelated to the the overlapping violation (restricted support) in [1]. **In our proof,** we show that the density ratio $\frac{P_{y\mid x}}{Q_{y\mid x}}$ between distributions with different ATEs escapes a bounded interval determined by the selection overlap $d$ and propensity bound $c$. **This argument is completely novel.**
>
>   (b) **When selection is deterministic,** it is not equivalent to overlap violation. Overlap violation is a property of the true underlying distribution, while deterministic selection is a problem of the data collection process. **Our key insight is to reduce the selection bias problem into a density extrapolation problem.** This coincides with the Taylor extrapolation result in truncated statistics [2], which [1] also uses in their paper. As the two problems are different, the reduction argument is different.
>
> - **Implementable estimators.** A key practical contribution of ours that distinguishes us from [1] is that we provide concrete, implementable estimation procedures (Algorithm 1). [1] only provides theoretical guarantee, but no practical estimator is presented. Our proposed MLE$+\hat{\beta}$ and Score Matching (SM$+\hat{\beta}$) estimator is validated through synthetic and semi-synthetic experiments.
>
> **Sparked by the feedback we further justify the novelty and general applicability of** the proposed framework. We provide extra results in two novel settings: Gnome-Wide-Association Studies (GWAS) and Mendelian Randomization Studies (MR). Due to space, **we point the reviewer to R4(4sDt) Q2 for the results** and will provide the theory in the revision.
>
> **Q3 & Q4: Typos and notational inconsistencies**
>
> We define observational study as $\mathcal{P}=X, Y(T), T$, and the finite sample dataset as $\mathcal{D}=\{X\_i, Y\_i, T\_i\}\_{i=1}^N$. Definition B.2, B.3 and Theorem 3.1 are defined over the observational studies $\mathcal{P}$. We define $\hat{\beta}(x,y,t)$ as the estimated $\alpha(x, y,t):=\frac{P\_{s\mid x, y, t}}{P\_s(S=1)}$. In Prop 3.5, the parameter $d$ indicates that $\forall d \in (0, 1)$, the distribution classes can satisfy Condition 1. We thank the reviewer for pointing out the inconsistencies and will correct them all in the revision.
>
> **Q5: Unclear about the experiments and the estimation procedure.**
>
> We kindly refer the reviewer to **R4(4sDt) Q2 and R3(4rWp) Q4** for more experiment analysis and apologize for this inconvenience.
>
> [1] Cai et al., 2025
> [2] Daskalakis et al., 2021 (Please refer to the main paper for full citation)

---

> > ### Author Rebuttal · Reviewer_kGhY · 2026-04-05
> >
> > I thank the authors for their response.
> > Regarding the response to Q1 - I am not entirely convinced. I think it is irrelevant that Condition 1 specifies a property of the distribution class because my claim (which the authors agree to) is that that property is just saying that if the ATE is different for two distributions in the distribution class, then the distributions are different. It is possible for Theorem 3.1's proof technique to not be immediate from the definition for my claim to still hold.
> > Regarding response to Q2- Thank you for explaining the differences.
> > Regarding response to Q5 - I don't think my concern is resolved by additional experiments. I have a specific concern about the results in the paper. From the response to the other reviewers I don't think that got addressed?
> >
> > In general, I also think that the proposed new experiments are substantial revisions to the current draft that I don't expect myself to be able to evaluate enough, given the space constraints, to recommend acceptance. So, I will keep my score unchanged.

---

> > > ### Author Response · Authors · 2026-04-06
> > >
> > > Thank you for continuing to engage with us and for the thoughtful follow-up. We appreciate the time you have taken to evaluate our responses, and we address your remaining concerns below.
> > >
> > > **Regarding Q1 (Condition 1 and Theorem 3.1):**
> > >
> > > We agree with your observation that Condition 1 is, at its core, a contrapositive statement of identifiability restricted to the distribution class triple. We thank you for pointing it out to help us clarify the role of each result in our paper. We will revise the manuscript to make it explicit that the primary contributions lie in Propositions 3.3 and 3.5, which provide concrete, testable instantiations of the condition for specific distribution families. We are happy to rename Theorem 3.1 (e.g., as an alternative "Definition (at a distribution class level)" or "Proposition") to more accurately reflect its role.
> > >
> > > That said, we would like to note that **Theorem 3.1 still serves an important structural purpose in the paper: it provides a unified language** that allows us to cast all new results (Propositions 3.3 and 3.5, whose proofs are not trivial) and all existing graphical identifiability results (Corollaries 3.9–3.13) into a single framework and to state new identifiability guarantees in a coherent manner. We will add a discussion paragraph in the revised text to this framing clearly without overclaiming the theorem's novelty.
> > >
> > > **Regarding Q5 (Experiments and estimation procedure):**
> > >
> > > We apologize for not addressing your specific concern more directly in our previous response because we are short of space. The relatively good performance of the Polynomial estimator in our main synthetic experiments is because we use polynomial functions (Appendix D) for the true data generating process (in both synthetic and semi-synthetic setting). This gives the Polynomial baseline a structural advantage.
> > >
> > > Therefore, we conduct extra experiments where we use non-polynomial structural equations (sin and log functions, as also in response to R4(4sDt)Q2), the Polynomial estimator degrades substantially, while MLE+β and SM+β maintain robust performance. Specifically, we use $Y = 2X\sin(2X)+ T(X^2+0.1X^4)+\epsilon$ (Sin) and $Y=X\log (X+4)+ T(X^2 \log (2X)+0.1X^4)+\epsilon$ (Log). We apply the same deterministic and nondeterministic selection mechanisms as in the main experiments. **This demonstrates that our proposed methods generalize more reliably across functional forms**.
> > >
> > > **Table (Varying function): Mean Error (std)**
> > >
> > > | **Sin**     | **Log**         |                 |
> > > | ----------- | --------------- | --------------- |
> > > | AIPW Oracle | 0.10 (0.06)     | 0.04 (0.05)     |
> > > | AIPW        | 3.61 (0.08)     | 3.85 (0.11)     |
> > > | TMLE        | 3.58 (0.07)     | 3.70 (0.12)     |
> > > | Heckman     | 4.25 (0.09)     | 4.26 (0.08)     |
> > > | IPW         | 3.62 (0.08)     | 4.34 (0.07)     |
> > > | Polynomial  | 3.66 (0.15)     | -2.58 (0.39)    |
> > > | MLE         | 2.04 (0.11)     | -3.71 (0.32)    |
> > > | MLE-\beta   | 3.63 (1.76)     | 1.66 (3.24)     |
> > > | SM          | 2.87 (1.11)     | -1.65 (0.84)    |
> > > | SM-\beta    | **1.28** (1.49) | **0.31** (3.47) |
> > >
> > > Regarding the semi-synthetic experiments on the All of Us dataset, we acknowledge the higher variance of MLE+β and SM+β. As noted in the paper, this dataset presents particularly challenging conditions (propensity scores around 0.05), leading to weak overlap. Despite the increased variance, our methods consistently reduce the bias relative to the baseline which does not take into account selection bias (note that the Polynomial estimator can still adjust for deterministic selection, as stated in Table 2). We will add a more thorough discussion of these results in the revision.
> > >
> > > **Regarding the additional GWAS and Mendelian Randomization experiments:** we understand your concern about adding substantial new material during the rebuttal period. These experiments were included in response to other reviewers who ask for real-world applications. We mention them here to illustrate that the framework introduced naturally extends to additional settings beyond those in the current paper (via Theorem 3.1 which provides a unifying language for us to characterize selection bias). The new experiments are supplementary evidence of the framework's generality, not a revision of the paper's central claims.
> > >
> > > We sincerely appreciate your feedback as it has been helpful for us to improve the presentation and clarify the contributions, and we will incorporate all of these improvements in the revision. Please let us know if you have further concerns. Thank you!

---

### Decision · Program_Chairs · 2026-04-30

**Decision:**

Accept (regular)

**Comment:**

This paper establishes new identifiability results for average treatment effects in the presence of selection bias, which extend the graphical identification criteria in the literature. It also proposes estimation procedures and demonstrates their empirical performance.

There is consensus among the reviewers that the paper makes original and potentially useful contributions to an important problem. One reviewer expressed concern about the apparent triviality of Condition 1 as a necessary and sufficient criterion for ATE identifiability, as well as about the precise interpretation and implications of the experimental results, and on this basis recommended against acceptance. However, the authors’ responses, especially in the second round, appear to have adequately addressed these concerns.

The other three reviewers are generally favorable, though they also raise critical points regarding the positioning and presentation of the results, the scope of applicability, the limitations of the empirical evaluation, and the strength of the underlying assumptions. In their rebuttal, the authors provided detailed responses to these comments, which appear to have satisfied the reviewers.

My own assessment aligns with the three positive reviews: the paper delivers strong and insightful theoretical results, along with useful methodological contributions toward addressing a major challenge in causal inference.